# MambaAD: Exploring State Space Models for Multi-class Unsupervised Anomaly Detection

**Haoyang He**[1*]  **Yuhu Bai**[1*]  **Jiangning Zhang**[2†]  **Qingdong He**[2]  **Hongxu Chen**[1]
**Zhenye Gan**[2]  **Chengjie Wang**[2]  **Xiangtai Li**[3]  **Guanzhong Tian**[1]  **Lei Xie**[1†]
[1]Zhejiang University   [2]Youtu Lab, Tencent   [3]Nanyang Technological University
{haoyanghe,yhbai,186368,chenhongxu,gztian}@zju.edu.cn,
{yingcaihe,wingzygan,jasoncjwang}@tencent.com,
xiangtai94@gmail.com, leix@iipc.zju.edu.cn

## Abstract

Recent advancements in anomaly detection have seen the efficacy of CNN- and transformer-based approaches. However, CNNs struggle with long-range dependencies, while transformers are burdened by quadratic computational complexity. Mamba-based models, with their superior long-range modeling and linear efficiency, have garnered substantial attention. This study pioneers the application of Mamba to multi-class unsupervised anomaly detection, presenting MambaAD, which consists of a pre-trained encoder and a Mamba decoder featuring (Locality-Enhanced State Space) LSS modules at multi-scales. The proposed LSS module, integrating parallel cascaded (Hybrid State Space) HSS blocks and multi-kernel convolutions operations, effectively captures both long-range and local information. The HSS block, utilizing (Hybrid Scanning) HS encoders, encodes feature maps into five scanning methods and eight directions, thereby strengthening global connections through the (State Space Model) SSM. The use of Hilbert scanning and eight directions significantly improves feature sequence modeling. Comprehensive experiments on six diverse anomaly detection datasets and seven metrics demonstrate state-of-the-art performance, substantiating the method's effectiveness. The code and models are available at https://lewandofskee.github.io/projects/MambaAD.

## 1  Introduction

The advent of smart manufacturing has markedly increased the importance of industrial visual Anomaly Detection (AD) in production processes. This technology promises to enhance efficiency, diminish the costs of manual inspections, and elevate product quality along with the stability of production lines. Presently, most methods predominantly utilize a single-class setting [12, 30, 55], where a separate model is trained and tested for each class, leading to considerable increases in training and memory usage. Despite recent progress in introducing multi-class AD techniques [47, 18], there is still significant potential for advancement in terms of both accuracy and efficiency trade-off.

The current unsupervised anomaly detection algorithms can be broadly categorized into three approaches [50, 9, 51]: *Embedding-based* [36, 11, 4, 12, 8], *Synthesizing-based* [48, 24, 55, 30], and *Reconstruction-based* [27, 18, 6]. Despite the promising results of both Synthesizing and Embedding-based methods in AD, these approaches often require extensive design and inflexible frameworks. Reconstruction-based methods, such as RD4AD [12] and UniAD [47], exhibit superior performance

---

[*]Equal contributions.

[†]Corresponding author.

38th Conference on Neural Information Processing Systems (NeurIPS 2024).

and better scalability. RD4AD, as depicted in Fig. 1 (a), employs a pre-trained teacher-student model, comparing anomalies across multi-scale feature levels. While CNN-based RD4AD captures local context effectively, *it lacks the ability to establish long-range dependencies.* UniAD, the first multi-class AD algorithm, relies on a pre-trained encoder and transformer decoder architecture as illustrated in Fig. 1 (b). Despite their superior global modeling capabilities, *transformers are hampered by quadratic computational complexity, which confines UniAD to anomaly detection on the smallest feature maps, potentially impacting its performance.*

Recently, Mamba [15] has demonstrated exceptional performance in large language models, offering significantly lower linear complexity compared to transformers while maintaining comparable effectiveness. Numerous recent studies have incorporated Mamba into the visual domain, sparking a surge of research [29, 57, 40, 22, 37, 43]. This paper pioneers the application of Mamba into the anomaly detection area, introducing MambaAD, as illustrated in Fig. 1 (c). *MambaAD combines global and local modeling capabilities, leveraging its linear complexity to compute anomaly maps across multiple scales.* Notably, it boasts a lower parameter count and computational demand, making it well-suited for practical applications.

Specifically, MambaAD employs a pyramid-structured auto-encoder to reconstruct multi-scale features, utilizing a pre-trained encoder and a novel decoder based on the Mamba architecture. This Mamba-based decoder consists of Locality-Enhanced State Space (LSS) modules at varying scales and quantities. Each LSS module comprises two components: a series of Hybrid State Space (HSS) blocks for global information capture and parallel multi-kernel convolution operations for establishing local connections. The resulting output features integrate the global modeling capabilities of the mamba structure with the local correlation strengths of CNNs. The proposed HSS module investigates five distinct scanning methods and eight scanning directions, with the (Hybrid Scanning) HS encoder and decoder encoding and decoding feature maps into sequences of various scanning methods and directions, respectively. The HSS module enhances the global receptive field across multiple directions, and its use of the Hilbert scanning method [19, 23] is particularly suited to the central concentration of industrial product features. By computing and summing anomaly maps across different feature map scales, MambaAD achieves SoTA performance on several representative AD datasets with seven different metrics for both image- and pixel-level while maintaining a low model parameter count and computational complexity. Our contributions are as follows:

- We introduce MambaAD, which innovatively applies the Mamba framework to address multi-class unsupervised anomaly detection tasks. This approach enables multi-scale training and inference with minimal model parameters and computational complexity.

- We design a Locality-Enhanced State Space (LSS) module, comprising cascaded Mamba-based blocks and parallel multi-kernel convolutions, extracts both global feature correlations and local information associations, achieving a unified model of global and local patterns.

- We have explored a Hybrid State Space (HSS) block, encompassing five methods and eight multi-directional scans, to enhance the global modeling capabilities for complex anomaly detection images across various categories and morphologies.

- We demonstrate the superiority and efficiency of MambaAD in multi-class anomaly detection tasks, achieving SoTA results on *six* distinct AD datasets with *seven* metrics while maintaining remarkably low model parameters and computational complexity.

## 2 Related Work

### 2.1 Unsupervised Anomaly Detection

**Unsupervised Anomaly Detection.** Existing AD methods can be mainly categorized into three types: *1) Embedding-based methods* focus on encoding RGB images into multi-channel features [36, 11, 4, 12, 10, 38]. These methods typically employ networks per-trained on ImageNet [13]. PatchCore [36] extracts nominal patch features with a memory bank for measuring the Mahalanobis distance. [4] is based on a student-teacher framework where student networks are trained to regress the output of a teacher network. However, the datasets used for these pre-trained models have a significant distribution gap compared to industrial images. *2) Synthesizing-based methods* synthesize anomalies on normal images [48, 24, 39, 20]. The pseudo-anomalies in DREAM [48] are generated

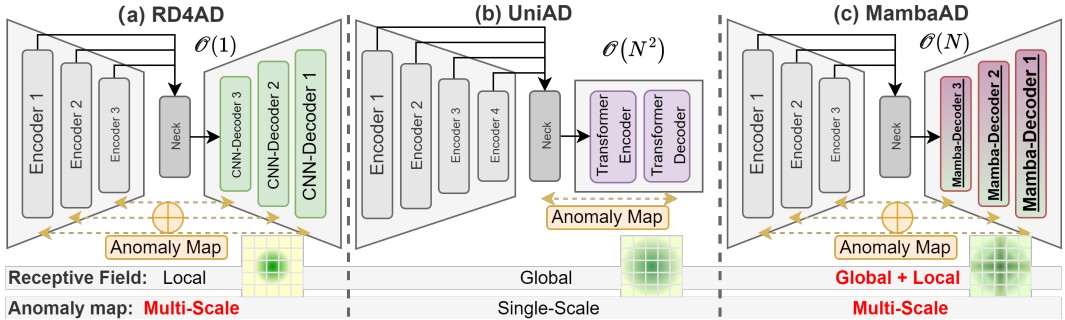

Figure 1: Compared with (a) local CNN-based RD4AD [12] and (b) global Transformer-based UniAD [47], ours MambaAD with linear complexity is capable of integrating the advantages of both global and local modeling, and multi-scale features endow it with more refined prediction accuracy.

utilizing Perlin noise and texture images. DREAM, taking anomaly mask as output, consists of a reconstruction network and a discriminative network. Despite the decent performance of such methods, the synthesized anomalies still have a certain gap compared to real-world anomalies. *3) Reconstruction-based methods* [12, 27, 52, 7] typically focus on self-training encoders and decoders to reconstruct images, reducing reliance on pre-trained models. Autoencoder [35], Transformers[33], Generative Adversarial Networks (GANs) [27, 46] and diffusion models [18, 45] can serve as the backbone for reconstruction networks in anomaly detection. While the model's generalization can occasionally lead to inaccuracies in pinpointing anomalous areas.

**Multi-class Anomaly Detection.** Most current works are trained individually on separate categories, which leads to increased time and memory consumption as the number of categories grows, and they are uncongenial to situations with large intra-class diversity. Recently, to address these issues, multi-class unsupervised anomaly detection (MUAD) methods have attracted a lot of interest. UniAD [47] firstly crafts a unified reconstruction framework for anomaly detection. DiAD [18] investigates an anomaly detection framework based on diffusion models, introducing a semantic-guided network to ensure the consistency of reconstructed image semantics.ViTAD [50] further explores the effectiveness of vanilla Vision Transformer (ViT) on multi-class anomaly detection.

## 2.2 State Space Models

State space models (SSMs) [17, 16, 41, 32, 14] have gained considerable attention due to their efficacy in handling long language sequence modeling. Specifically, structure state-space sequence (S4) [16] efficiently models long-range dependencies (LRDs) through parameterization with a diagonal structure, addressing computational bottlenecks encountered in previous works. Building upon S4, numerous models have been proposed, including S5 [41], H3 [14], and notably, Mamba [15]. Mamba introduces a data-dependent selection mechanism into S4, which provides a novel paradigm distinct from CNNs or Transformers, maintaining linear scalability of long sequences processing.

The tremendous potential of Mamba has sparked a series of excellent works [29, 57, 37, 21, 22, 40, 44, 43, 31, 54, 26] in the vision domain. Vmamba [29] proposes a cross-scan module (CSM) to tackle the direction sensitivity issue between non-causal 2D images and ordered 1D sequences. Moreover, Mamba has found extensive use in the domain of medical image segmentation [37, 28, 44, 43, 25], incorporating Mamba blocks to UNet-like architecture to achieve task-specific architecture. VL-Mamba [34] and Cobra [56] explore the potential of SSMs in multimodal large language models. Besides, ZigMa [21] addresses the spatial continuity in the scan strategy, and it incorporates Mamba into the Stochastic Interpolation framework [1].

In this work, we develop MambaAD to exploit Mamba's long-range modeling capacity and linear computational efficiency for multi-class unsupervised anomaly detection. This approach innovatively combines SSM's global modeling capabilities with CNNs' detailed local modeling prowess.

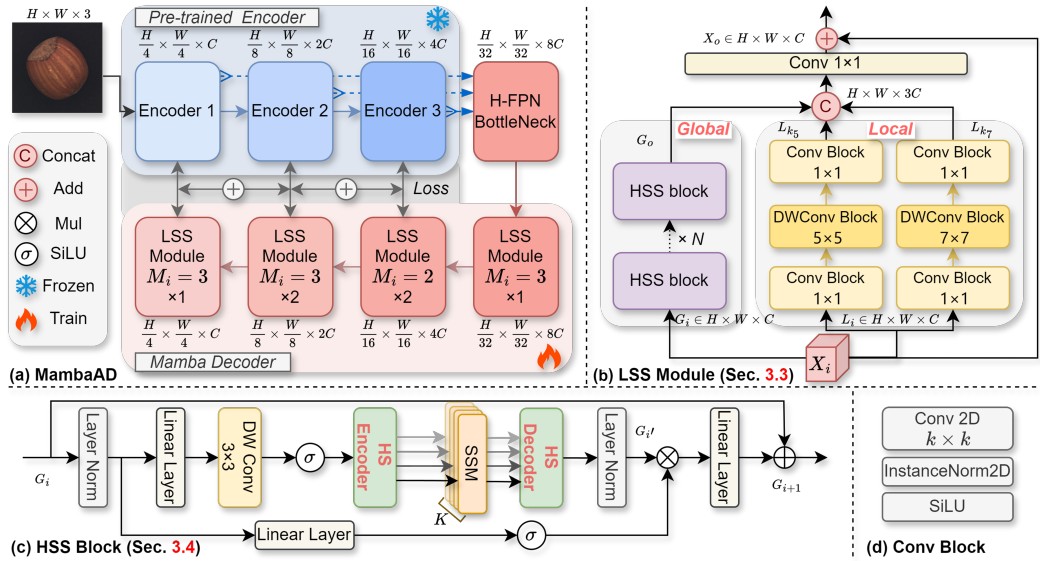

Figure 2: **Overview of the proposed MambaAD**, which employs pyramidal auto-encoder framework to reconstruct multi-scale features by the proposed efficient and effective Locality-Enhanced State Space (LSS) module. Specifically, each LSS consists of: *1)* cascaded Hybrid State Space (HSS) blocks to capture global interaction; and *2)* parallel multi-kernel convolution operations to replenish local information. Aggregated multi-scale reconstruction error serves as the anomaly map for inference.

## 3 Method

### 3.1 Preliminaries

State Space Models [16], inspired by control systems, map a one-dimensional stimulation $x(t) \in \mathbb{R}^L$ to response $y(t) \in \mathbb{R}^L$ through a hidden state $h(t) \in \mathbb{R}^N$, which are formulated as linear ordinary differential equations (ODEs):

$$h'(t) = \mathbf{A}h(t) + \mathbf{B}x(t), \quad y(t) = \mathbf{C}h(t), \tag{1}$$

where the state transition matrix $\mathbf{A} \in \mathbb{R}^{N \times N}$, $\mathbf{B} \in \mathbb{R}^{N \times 1}$ and $\mathbf{C} \in \mathbb{R}^{1 \times N}$ for a state size $N$.

S4 [16] and Mamba [15] utilize zero-order hold with a timescale parameter $\Delta$ to transform the continuous parameters $\mathbf{A}$ and $\mathbf{B}$ from the continuous system into the discrete parameters $\overline{\mathbf{A}}$ and $\overline{\mathbf{B}}$:

$$\overline{\mathbf{A}} = \exp(\mathbf{\Delta A}), \quad \overline{\mathbf{B}} = (\mathbf{\Delta A})^{-1}(\exp(\mathbf{\Delta A}) - \mathbf{I}) \cdot \mathbf{\Delta B}. \tag{2}$$

After the discretization, the discretized model formulation can be represented as:

$$h_t = \overline{\mathbf{A}}h_{t-1} + \overline{\mathbf{B}}x_t, \quad y_t = \mathbf{C}h_t. \tag{3}$$

At last, from the perspective of global convolution, the output can be defined as:

$$\overline{\mathbf{K}} = (\mathbf{C}\overline{\mathbf{B}}, \mathbf{C}\overline{\mathbf{A}}\overline{\mathbf{B}}, \dots, \mathbf{C}\overline{\mathbf{A}}^{L-1}\overline{\mathbf{B}}), \quad \mathbf{y} = \mathbf{x} * \overline{\mathbf{K}}, \tag{4}$$

where $*$ represents convolution operation, $L$ is the length of sequence $x$, and $\overline{\mathbf{K}} \in \mathbb{R}^L$ is a structured convolutional kernel.

### 3.2 MambaAD

The MambaAD framework is proposed for multi-class anomaly detection as illustrated in Fig. 2(a). It consists of three main components: a pre-trained CNN-based encoder, a Half-FPN bottleneck, and a Mamba-based decoder. During training, the encoder extracts feature maps at three different

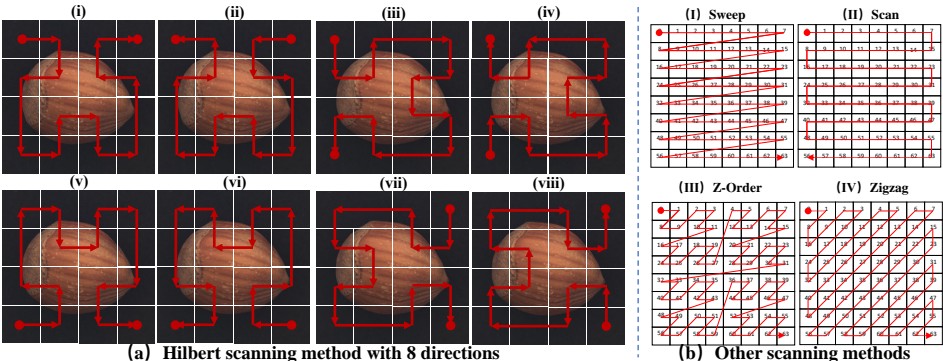

Figure 3: Hybrid Scanning directions and methods. *(a)* The Hilbert scanning method with 8 scanning directions is used for HS Encoder and Decoder. *(b)* The other four scanning methods for comparison.

scales and inputs them into the H-FPN bottleneck for fusion. The fused output is then fed into the Mamba Decoder with a depth configuration of [3,4,6,3]. The final loss function is the sum of the mean squared error (MSE) computed across feature maps at three scales. Within the Mamba Decoder, we introduce the Locality-Enhanced State Space (LSS) module. The LSS can be configured with different stages $M_i$, where each stage represents the number $N$ of Hybrid State Space (HSS) blocks within the module. In this experiment, we employ LSS with $M_i = 3$ and $M_i = 2$ as the primary modules. The LSS module processes the input $X_i$ through HSS blocks that capture global information and through two different scales of Depth-Wise Convolution (DWConv) layers that capture local information. The original input feature dimension is restored through concatenation and convolution operations. The proposed HSS block features a Hybrid Scanning (HS) Encoder and an HS Decoder, which accommodates five distinct scanning methods and eight scanning directions.

### 3.3 Locality-Enhanced State Space Module

Transformers excel in global modeling and capturing long-range dependencies but tend to overlook local semantic information and exhibit high computational complexity when processing high-resolution features. Conversely, CNNs effectively model local semantics by capturing information from adjacent positions but lack long-range modeling capabilities. To address these limitations, we propose the LSS module in Fig. 2 (b), which incorporates both Mamba-based cascaded HSS blocks for global modeling and parallel multi-kernel depth-wise convolution operations for local information capture.

Specifically, for an input feature $X_i \in \mathbb{R}^{H \times W \times C}$, global features $G_i \in \mathbb{R}^{H \times W \times C}$ enter the HSS blocks while local features $L_i \in \mathbb{R}^{H \times W \times C}$ proceed through a convolutional network. The global features $G_i$ pass through a series of $N$ HSS blocks to obtain global information features $G_o$.

$$G_o = \mathbf{HSS_n}(...(\mathbf{HSS_2}(\mathbf{HSS_1}(G_i)))), \tag{5}$$

where $n \in N$ is the number of HSS blocks. In this study, we primarily use $N = 2$ and $N = 3$, with further ablation experiments presented in Sec. 4.3.

Local features $L_i$ are processed by two parallel DWConv blocks, each comprising a $1 \times 1$ Conv block, an $k \times k$ DWConv block, and another $1 \times 1$ Conv block.

$$L_m = \mathbf{ConvB_{1 \times 1}}(\mathbf{DWConvB_{k \times k}}(\mathbf{ConvB_{1 \times 1}}(L_i)), \tag{6}$$

where $k$ is the kernel size for the DWConv. $k = 5$ and $k = 7$ are used in this experiment with further ablations in Sec. 4.3. Each convolutional module includes a Conv 2D layer, an Instance Norm 2D layer, and a SiLU as illustrated in Fig. 2 (d). Local and global features are aggregated by concatenation along the channel dimension. The final output $X_o$ of this block is obtained by a $1 \times 1$ 2D convolution to restore the channel count to match that of the input and a residual connection.

$$X_o = \mathbf{Conv2D_{1 \times 1}}(\mathbf{Concat}(G_o, L_{k_5}, L_{k_7})) + X_i. \tag{7}$$

### 3.4 Hybrid State Space Block

Following the method outlined in [29, 37], the HSS block is designed for hybrid-method and hybrid-directional scanning and fusion, as depicted in Fig. 2 (c). The HSS block primarily comprises Layer

Normalization (LN), Linear Layer, depth-wise convolution, SiLU activation, Hybrid Scanning (HS) encoder $\mathcal{E}_{HS}$, State Space Models (SSMs), HS decoder $\mathcal{D}_{HS}$, and residual connections.

$$G'_i = \mathbf{LN}(\mathcal{D}_{HS}(\mathbf{SSMs}(\mathcal{E}_{HS}(\sigma(\mathbf{DWConv_{3\times3}}(\mathbf{Linear}(\mathbf{LN}(G_i))))))))),$$
$$G_{i+1} = \mathbf{Linear}(G'_i \cdot \sigma(\mathbf{Linear}(\mathbf{LN}(G_i)))) + G_i \tag{8}$$

**Hybrid Scanning methods.** Inspired by space-filling curves [49, 53], as shown in Fig. 3, this study explores five different scanning methods: (I) Sweep, (II) Scan, (III) Z-order, (IV) Zigzag, and (V) Hilbert, to assess their impact on the SSM's modeling capabilities. The Hilbert scanning method is ultimately selected for its superior encoding and modeling of local and global information within feature sequences, particularly in mitigating the challenges of modeling long-range dependencies. Further experimental results will be presented in the ablation study. Assuming $A$ is a matrix, $A^T$ is the transpose of $A$, $A^{lr}$ is the left-right reversal of $A$, $A^{ud}$ is the up-down reversal of $A$. The Hilbert curve can be obtained by an n-order Hilbert matrix:

$$H_{n+1} = \begin{cases} \begin{pmatrix} H_n & 4^n E_n + H_n^T \\ (4^{n+1}+1)E_n - H_n^{ud} & (3\times4^n+1)E_n - (H_n^{lr})^T \end{pmatrix} & \text{, if } n \text{ is even,} \\ \begin{pmatrix} H_n & (4^{n+1}+1)E_n - H_n^{lr} \\ 4^n E_n + H_n^T & (3\times4^n+1)E_n - (H_n^T)^{lr} \end{pmatrix} & \text{, if } n \text{ is odd,} \end{cases} \tag{9}$$

where $H_1 = \begin{pmatrix} 1 & 2 \\ 4 & 3 \end{pmatrix}$ and $E_n$ is all-one matrix for n-order.

**Hybrid Scanning directions.** Following the setup of previous scanning directions, this study supports eight Hilbert-based scanning directions: (i) forward, (ii) reverse, (iii) width-height (wh) forward, (iv) wh reverse, (v) rotated 90 degrees forward, (vi) rotated 90 degrees reverse, (vii) wh rotated 90 degrees forward, and (viii) wh rotated 90 degrees reverse, as illustrated in Fig. 3 (a). Multiple scanning directions enhance the encoding and modeling capabilities of feature sequences, enabling the handling of various types of anomalous features with further ablations in Sec. 4.3.

The HS encoder aims to combine and encode input features according to different scanning methods and directions before feeding them into the SSM to enhance the global modeling capacity of the feature vectors. The HS decoder then decodes the feature vectors output by the SSM back to the original input feature orientation, with the final output obtained by summation.

## 4 Experiments

### 4.1 Setups: Datasets, Metrics, and Details

**MVTec-AD** [3] encompasses a diverse collection of 5 types of textures and 10 types of objects, 5,354 high-resolution images in total. 3,629 normal images are designated for training. The remaining 1,725 images are reserved for testing and include both normal and abnormal samples.

**VisA** [58] features 12 different objects, incorporating three diverse types: complex structures, multiple instances, and single instances. It consists of a total of 10,821 images, of which 9,621 are normal samples, and 1,200 are anomaly samples.

**Real-IAD** [42] includes objects from 30 distinct categories, with a collection of 150K high-resolution images, making it larger than previous anomaly detection datasets. It consists of 99,721 normal images and 51,329 anomaly images.

More results on MVTec-3D [5], as well as newly proposed Uni-Medical [50, 2] and COCO-AD [52] datasets, can be viewed in Appendix 5.

**Metrics.** For anomaly detection and segmentation, we report Area Under the Receiver Operating Characteristic Curve (AU-ROC), Average Precision [48] (AP) and F1-score-max [58] (F1_max). Additionally, for anomaly segmentation, we also report Area Under the Per-Region-Overlap [4] (AU-PRO). We further calculate the mean value of the above seven evaluation metrics (denoted as $mAD$) to represent a model's comprehensive capability [50].

**Implementation Details.** All input images are resized to a uniform size of $256 \times 256$ without additional augmentation for consistency. A pre-trained ResNet34 acts as the feature extractor, while a

Table 1: Quantitative Results on different AD datasets for multi-class setting.

| Dataset | Method | Image-level | | | Pixel-level | | | | mAD |
|---|---|---|---|---|---|---|---|---|---|
| | | AU-ROC | AP | F1_max | AU-ROC | AP | F1_max | AU-PRO | |
| MVTec-AD [3] | RD4AD [12] | 94.6 | 96.5 | 95.2 | 96.1 | 48.6 | 53.8 | 91.1 | 82.3 |
| | UniAD [47] | 96.5 | 98.8 | 96.2 | 96.8 | 43.4 | 49.5 | 90.7 | 81.7 |
| | SimpleNet [30] | 95.3 | 98.4 | 95.8 | 96.9 | 45.9 | 49.7 | 86.5 | 81.2 |
| | DeSTSeg [55] | 89.2 | 95.5 | 91.6 | 93.1 | 54.3 | 50.9 | 64.8 | 77.1 |
| | DiAD [18] | 97.2 | 99.0 | 96.5 | 96.8 | 52.6 | 55.5 | 90.7 | 84.0 |
| | MambaAD (Ours) | **98.6 ± 0.3** | **99.6 ± 0.2** | **97.8 ± 0.4** | **97.7 ± 0.4** | **56.3 ± 0.7** | **59.2 ± 0.6** | **93.1 ± 0.3** | **86.0 ± 0.3** |
| VisA [58] | RD4AD [12] | 92.4 | 92.4 | 89.6 | 98.1 | 38.0 | 42.6 | 91.8 | 77.8 |
| | UniAD [47] | 88.8 | 90.8 | 85.8 | 98.3 | 33.7 | 39.0 | 85.5 | 74.6 |
| | SimpleNet [30] | 87.2 | 87.0 | 81.8 | 96.8 | 34.7 | 37.8 | 81.4 | 72.4 |
| | DeSTSeg [55] | 88.9 | 89.0 | 85.2 | 96.1 | 39.6 | 43.4 | 67.4 | 72.8 |
| | DiAD [18] | 86.8 | 88.3 | 85.1 | 96.0 | 26.1 | 33.0 | 75.2 | 70.1 |
| | MambaAD (Ours) | **94.3 ± 0.4** | **94.5 ± 0.5** | **89.4 ± 0.6** | **98.5 ± 0.3** | 39.4 ± 1.1 | **44.0 ± 1.3** | 91.0 ± 0.9 | **78.7 ± 0.5** |
| Real-IAD [42] | RD4AD [12] | 82.4 | 79.0 | 73.9 | 97.3 | 25.0 | 32.7 | 89.6 | 68.6 |
| | UniAD [47] | 83.0 | 80.9 | 74.3 | 97.3 | 21.1 | 29.2 | 86.7 | 67.5 |
| | SimpleNet [30] | 57.2 | 53.4 | 61.5 | 75.7 | 2.8 | 6.5 | 39.0 | 42.3 |
| | DeSTSeg [55] | 82.3 | 79.2 | 73.2 | 94.6 | 37.9 | 41.7 | 40.6 | 64.2 |
| | DiAD [18] | 75.6 | 66.4 | 69.9 | 88.0 | 2.9 | 7.1 | 58.1 | 52.6 |
| | MambaAD (Ours) | **86.3 ± 0.4** | **84.6 ± 0.3** | **77.0 ± 0.4** | **98.5 ± 0.1** | 33.0 ± 0.6 | 38.7 ± 0.6 | **90.5 ± 0.3** | **72.7 ± 0.4** |

Mamba decoder of equivalent depth [3,4,6,3] to ResNet34 serves as the student model for training. In the Mamba decoder, the number of cascaded HSS blocks in the second LSS module is set to 2, while all other LSS modules employ 3 cascaded HSS blocks. This experiment employs the Hilbert scanning technique, utilizing eight distinct scanning directions. The AdamW optimizer is employed with a learning rate of $0.005$ and a decay rate of $1 \times 10^{-4}$. The model undergoes a training period of 500 epochs for the multi-class setting, conducted on a single NVIDIA TESLA V100 32GB GPU. During training, the sum of MSE across different scales is employed as the loss function. In the testing phase, the sum of cosine similarities at various scales is utilized as the anomaly maps.

## 4.2 Comparison with SoTAs on Different AD datasets

We compared our method with current SoTA methods on a range of datasets utilizing both image-level and pixel-level metrics (*c.f.*, Sec. 4.1). This paper primarily compares with UniAD [47] and DiAD [18] dedicated to MUAD. In addition, we also compare our MambaAD with Reconstruction-based RD4AD [12] and Embedding-based DeSTSeg [55]/SimpleNet [30].

**Quantitative Results.** As shown in Tab. 1, on MVTec-AD dateset, our MambaAD outperforms all the comparative methods and reaches a new SoTA to **98.6**/**99.6**/**97.6** and **97.7**/**56.3**/**59.2**/**93.1** in multi-class anomaly detection and segmentation. Specifically, compared to DiAD [18], our proposed MambaAD shows an improvement of 1.4 ↑/0.6 ↑/1.3 ↑ at image-level and 0.9 ↑/3.7 ↑/3.7 ↑/2.4 ↑ at pixel-level. Notably, for overall metric mAD of a model, our MambaAD improves by 2.0 ↑, compared with SoTA DiAD. The VisA dataset is more complex and challenging, yet our method still

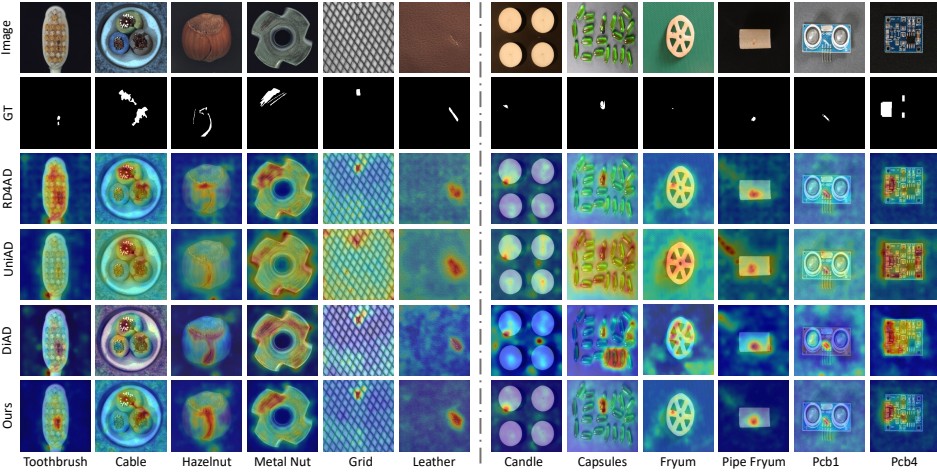

Figure 4: Qualitative visualization for pixel-level anomaly segmentation on MVTec and VisA datasets.

demonstrates excellent performance. As shown in Tab. 1, our MambaAD exceeds the performance of DiAD [18] by 7.5 ↑/6.2 ↑/4.3 ↑ at image-level and by 2.5 ↑/13.3 ↑/11.0 ↑/15.8 ↑. Meanwhile, we achieve an enhancement of 8.7 ↑ compared to advanced DiAD on the mAD metric. In addition, the SoTA results on Real-IAD datasets, shown in Tab. 1, illustrate the scalability, versatility, and efficacy of our method MambaAD. We also compared the results of MambaAD with state-of-the-art (SoTA) methods for single-class anomaly detection in Tabs. A13 to A16 in the Appendix. In the single-class tasks, we included a comparison with the PatchCore [36] method. On the MVTec-AD dataset, SimpleNet and PatchCore achieved the best results at the image level, while our method achieved the second-best results. At the pixel level, DeSTSeg achieved the best results in most metrics, whereas our method achieved the best results in AUPRO. For the VisA dataset, our method achieved the best results in two metrics each at both the image and pixel levels. This demonstrates that our multi-class anomaly detection method can also achieve optimal or near-optimal results in single-class tasks, indicating its effectiveness and robustness. In contrast, existing single-class SoTA methods like SimpleNet and DeSTSeg, although effective in single-class tasks, show a significant performance drop in multi-class tasks. The PatchCore method can only operate in single-class tasks; for multi-class tasks, it encounters issues such as GPU and memory overflow due to the need to store all features in the Memory Bank. In summary, MambaAD exhibits strong robustness and effectiveness. More detailed results for each category are presented in the Appendix.

**Qualitative Results.** We conducted qualitative experiments on MVTev-AD and VisA datasets that substantiated the accuracy of our method in anomaly segmentation. Fig. 4 demonstrates that our method possesses more precise anomaly segmentation capabilities. Compared to DiAD, our method delivers more accurate anomaly segmentation without significant anomaly segmentation bias.

### 4.3 Ablation and Analysis

**Components IncrementalAblations.** The ablation experiments for the proposed components are summarized in Tab. 2. Using the most basic decoder purely based on Mamba and employing only the simplest two-directional sweep scanning method, we achieve an mAD score of 82.1 on the MVTec-AD dataset. Subsequently, by incorporating the proposed LSS module, which integrates the global modeling capabilities of Mamba with the local modeling capabilities of CNNs, the mAD score improves by +2.8%. Finally, replacing the original scanning directions and methods with HSS, which combines features from different scanning directions and employs the Hilbert scanning method, better aligns with the data distribution in most industrial scenarios where objects are centrally located in the image. This results in an additional +1.1% point improvement in the mAD score. Overall, the proposed MambaAD achieves an mAD score of 86.0 on the MVTec-AD dataset and 78.9 on the VisA dataset, reaching the SoTA performance.

Table 2: Incremental Ablations.

| Basic Mamba | LSS | HSS | MVTec-AD | VisA |
|:---:|:---:|:---:|:---:|:---:|
| ✓ | | | 82.1 | 72.9 |
| ✓ | ✓ | | 84.9 | 78.0 |
| ✓ | ✓ | ✓ | **86.0** | **78.9** |

Table 3: Ablation Study on the LSS Module.

| Method | Params(M) | FLOPs(G) | MVTec-AD | VisA |
|:---:|:---:|:---:|:---:|:---:|
| Local | 13.0 | 5.0 | 81.7 | 72.5 |
| Global | 22.5 | 7.5 | 82.1 | 72.9 |
| Local + Global | 25.7 | 8.3 | **86.0** | **78.9** |

**Local/Global Branches of the LSS Module.** We conducted three ablation experiments to verify the impact of branches in the LSS module as shown in Tab. 3. The Local branch represents the use of only the parallel CNN branch, without the Mamba-based HSS branch. The Global branch represents the use of only the Mamba-based HSS branch, without the parallel CNN branch, making the decoder in this structure purely Mamba-based. Finally, Global+Local represents the proposed LSS structure used in MambaAD, which combines the serial Mamba-based HSS with parallel CNN branches of different kernel sizes. The experimental results are shown in the table below. The Local branch, which uses only CNNs, has the lowest parameter count and FLOPs but also the lowest mAD metric, indicating high efficiency but suboptimal accuracy. The Global method, based on the pure Mamba structure, consumes more parameters and FLOPs than the Local method but shows a significant improvement in performance (+2.7%). Finally, the combined Global+Local method, which is the LSS module used in MambaAD, achieves the best performance with a notable improvement of (+1.6%) over the individual methods.

**Effectiveness comparison of different pre-trained backbone and Mamba decoder depth.** First, we compared various pre-trained feature extraction networks, focusing on the popular ResNet series

Table 4: Ablation studies on the pre-trained backbone and Mamba decoder depth.

| Backbone | Decoder Depth | Image-level | | | Pixel-level | | | | Params(M) | FLOPs(G) |
|---|---|---|---|---|---|---|---|---|---|---|
| | | AU-ROC | AP | F1_max | AU-ROC | AP | F1_max | AU-PRO | | |
| ResNet18 | [2,2,2,2] | 96.7 | 98.6 | 95.8 | 95.7 | 47.9 | 52.4 | 89.1 | 14.6 | 4.3 |
| | [3,4,6,3] | 96.6 | 98.8 | 96.4 | 96.8 | 53.2 | 56.2 | 91.8 | 20.3 | 6.2 |
| ResNet34 | [2,2,2,2] | 98.0 | 99.3 | 97.0 | 97.6 | 55.4 | 58.2 | 92.7 | 20.0 | 6.5 |
| | [2,9,2,2] | 97.6 | 99.3 | 97.3 | 97.7 | 56.4 | 59.0 | 93.2 | 26.1 | 7.9 |
| | [3,4,6,3] | **98.6** | **99.6** | 97.8 | 97.7 | 56.3 | 59.2 | 93.1 | 25.7 | 8.3 |
| ResNet50 | [3,4,6,3] | 98.4 | 99.4 | 97.7 | 97.7 | 54.2 | 57.0 | 92.3 | 251.0 | 60.3 |
| WideResNet50 | [3,4,6,3] | **98.6** | 99.5 | **98.0** | 98.0 | 57.9 | 60.3 | 93.8 | 268.0 | 68.1 |

as shown in Tab. 4. When maintaining a consistent Mamba decoder depth, we observed that ResNet18 performed the poorest, despite its minimal model size and computational complexity. ResNet50, with approximately 8 times more parameters and computations. Although WideResNet surpassed ResNet34 in certain metrics, it required nearly 10 times the parameters and computational cost. Consequently, after considering all factors, we elected to use ResNet34 as the backbone feature extractor. Then, we examined the impact of different Mamba decoder depths while keeping the backbone network constant. The depths [2,2,2,2] and [3,4,6,3] corresponded to ResNet18 and ResNet34 depths, respectively, while [2,9,2,2] was the prevalent choice in other methods. Our experiments revealed that the [3,4,6,3] depth, despite a slight increase in parameters and computations, consistently outperformed the other configurations.

Table 5: Ablations on different scanning methods and directions.

| Index | HS Methods with Different Directions | | | | | Image-level | | | Pixel-level | | | |
|---|---|---|---|---|---|---|---|---|---|---|---|---|
| | Sweep | Scan | Zorder | Zigzag | Hilbert | AU-ROC | AP | F1_max | AU-ROC | AP | F1_max | AU-PRO |
| 1 | 8 | - | - | - | - | 98.1 | 99.4 | 97.2 | 97.5 | 56.8 | 58.8 | 92.9 |
| 2 | - | 8 | - | - | - | 98.0 | 99.4 | 97.2 | 97.6 | 56.6 | 59.0 | **93.4** |
| 3 | - | - | 8 | - | - | 98.1 | 99.4 | 97.4 | 97.6 | 56.6 | 59.0 | 93.0 |
| 4 | - | - | - | 8 | - | 98.2 | 99.4 | 97.6 | 97.6 | 56.3 | 58.8 | 93.1 |
| 5 | - | - | - | - | 2 | 97.9 | 99.3 | 97.1 | **97.7** | 56.5 | **59.2** | 93.1 |
| 6 | - | - | - | - | 4 | 98.0 | 99.4 | 97.0 | **97.7** | **56.9** | 59.1 | 93.2 |
| 7 | - | - | - | - | 8 | **98.6** | **99.6** | **97.8** | **97.7** | 56.3 | **59.2** | 93.1 |
| 8 | - | - | - | 4 | 4 | 96.8 | 99.0 | 97.0 | 97.4 | 54.4 | 57.0 | 92.8 |
| 9 | - | - | 4 | - | 4 | 97.5 | 99.2 | 97.4 | 97.5 | 55.0 | 57.4 | 93.1 |
| 10 | - | 4 | - | - | 4 | 97.4 | 99.1 | 96.8 | 97.5 | 55.5 | 57.9 | 93.3 |
| 11 | 4 | - | - | - | 4 | 98.0 | 99.3 | 97.4 | 97.6 | 56.2 | 58.5 | 93.3 |
| 12 | - | 2 | 2 | 2 | 2 | 97.5 | 99.2 | 97.1 | 97.5 | 55.4 | 57.9 | 92.9 |

**Efficiency comparison of different SoTA methods.** In Tab. 6, we compared our model with five SoTA methods in terms of model size and computational complexity. MambaAD exhibits a minimal increase in parameters compared to UniAD. However, MambaAD outperforms it by 4.3 ↑ on the comprehensive metric mAD. Moreover, MambaAD significantly outperforms these other approaches, demonstrating its effectiveness in model lightweight design while maintaining high performance. Particularly, our method MambaAD has achieved about 2.0 ↑ improvement with only *1/50 the parameters and flops of DiAD*.

Table 6: Efficiency comparison of SoTA methods.

| Method | Params(M) | FLOPs(G) | mAD |
|---|---|---|---|
| RD4AD[12] | 80.6 | 28.4 | 82.3 |
| UniAD [47] | **24.5** | **3.6** | 81.7 |
| DeSTSeg [55] | 35.2 | 122.7 | 81.2 |
| SimpleNet [30] | 72.8 | 16.1 | 77.1 |
| DiAD [18] | 1331.3 | 451.5 | 84.0 |
| MambaAD (Ours) | 25.7 | 8.3 | **86.0** |

**Effectiveness and efficiency comparison of different scanning methods and directions.** In the initial stage, we compared five distinct scanning methods with an 8-direction scan as shown in Tab. 5. The results indicated that the other four methods produced similar outcomes as *Index 1-4*, albeit marginally inferior to the Hilbert scanning technique at the image level. Subsequently, we examined the impact of varying scan directions. As the number of scan directions increased as *Index 5-7*, image-level metrics improved gradually, while pixel-level metrics remained consistent. This suggests that augmenting the number of scan directions enhances the global modeling capability of SSM, thereby decreasing the likelihood of image-level misclassification. Ultimately, our analysis revealed that combining various scanning techniques while maintaining a total of eight scan directions led to a decline in performance as as *Index 8-12*. This decrement could be attributed to the significant disparities among the scanning approaches employed. Consequently, we opted for the Hilbert scanning method, as it demonstrated better suitability for real-world industrial products.

**Effectiveness comparison of different LSS designs.** In Tab. 7, the design focuses on three distinct design directions: the number of $M_i$ in the LSS module, the configuration of parallel multi-kernel convolution modules, and the kernel size selection for Depth-Wise Convolutions (DWConv). Initially,

Table 7: Ablation studies on LSS module's designs.

| LSS Design | Local Conv | Kernel Size | Image-level | | | Pixel-level | | | |
|---|---|---|---|---|---|---|---|---|---|
| | | | AU-ROC | AP | F1_max | AU-ROC | AP | F1_max | AU-PRO |
| $M_i = 1$ | Only DWConv | k=1,3 | 94.6 | 98.0 | 96.2 | 96.7 | 49.5 | 52.4 | 91.3 |
| | | k=3,5 | 94.6 | 98.0 | 96.4 | 96.6 | 50.1 | 53.5 | 91.3 |
| | | k=5,7 | 96.5 | 98.9 | 96.7 | 96.6 | 51.0 | 53.8 | 91.4 |
| | | k=1,3,5 | 94.8 | 98.2 | 96.1 | 95.9 | 47.7 | 51.6 | 89.8 |
| | | k=3,5,7 | 96.5 | 98.8 | 96.5 | 97.2 | 51.6 | 55.2 | 92.2 |
| | | k=1,3,5,7 | 95.2 | 98.3 | 96.5 | 96.5 | 49.1 | 52.6 | 90.8 |
| | DWConv + $1 \times 1$ Conv | k=3,5 | 96.1 | 98.7 | 96.7 | 97.2 | 52.2 | 55.6 | 92.5 |
| | | k=5,7 | 95.8 | 98.6 | 96.3 | 97.3 | 51.7 | 54.5 | 92.0 |
| | | k=3,5,7 | 96.0 | 98.6 | 96.7 | 97.3 | 52.4 | 55.4 | 92.2 |
| $M_i = 1$, noskip | Only DWConv | k=3,5 | 97.3 | 99.1 | 97.0 | 96.9 | 53.4 | 55.9 | 91.5 |
| | | k=5,7 | 95.0 | 98.2 | 96.0 | 96.8 | 48.6 | 52.8 | 91.5 |
| | | k=3,5,7 | 94.6 | 98.2 | 95.9 | 97.1 | 52.3 | 55.4 | 92.3 |
| | DWConv + $1 \times 1$ Conv | k=3,5 | 95.4 | 98.4 | 96.6 | 97.0 | 51.6 | 54.4 | 91.7 |
| | | k=5,7 | 95.0 | 98.3 | 96.3 | 97.1 | 50.1 | 53.9 | 91.8 |
| | | k=3,5,7 | 96.0 | 98.7 | 96.3 | 97.2 | 52.3 | 55.4 | 92.2 |
| $M_i = 2, M_i = 3$ | Only DWConv | k=3,5 | 98.2 | 99.2 | 97.0 | 97.4 | 55.7 | 58.2 | 92.8 |
| | | k=5,7 | 98.2 | 99.4 | 97.4 | **97.7** | 56.1 | 59.0 | 92.9 |
| | | k=3,5,7 | 98.2 | 99.3 | 97.3 | 97.5 | 53.9 | 57.5 | 92.9 |
| | DWConv + $1 \times 1$ Conv | k=3,5 | **98.6** | 99.5 | 97.6 | 97.5 | 55.3 | 58.3 | 92.7 |
| | | k=5,7 | **98.6** | **99.6** | **97.8** | **97.7** | **56.3** | **59.2** | **93.1** |
| | | k=3,5,7 | 98.1 | 99.3 | 97.4 | 97.5 | 55.1 | 57.6 | 92.8 |

we compare scenarios where $M_i = 1$, meaning each LSS module contains a single HSS block, and we experiment with different kernel sizes for DWConv alone and configurations flanked by $1 \times 1$ convolutions. Subsequently, we contrast the results without residual connections against those with identical settings but with $M_i = 1$. Finally, we examine the outcomes when $M_i = 2$ and $M_i = 3$ under otherwise consistent settings. Analysis reveals that with $M_i = 1$, regardless of the presence of residual connections, the results are inferior to those with $M_i = 2$ and $M_i = 3$. Moreover, using only DWConv blocks with $M_i = 1$, a comparison of parallel depth-wise convolutions with varying kernel sizes indicates that smaller kernels, such as $k = 1$, significantly degrade performance. Therefore, subsequent comparative experiments focus on larger convolution kernels. In the absence of residual connections, some metrics may surpass those with residual connections, but the longer training time and difficulty in convergence preclude their use in further experiments. In configurations with $M_i = 2$ and $M_i = 3$, we find that DWConv blocks augmented with $1 \times 1$ convolutions exhibit superior performance. Additionally, kernels sized $k = 5$ and $k = 7$ are more suitable for extracting local features and establishing local information associations. Consequently, in this study, we opt for a quantity of HSS blocks with $M_i = 2$ and $M_i = 3$, and we employ parallel DWConv blocks with kernels sized $k = 5$ and $k = 7$, complemented by $1 \times 1$ convolutions before and after.

## 5 Conclusion

This paper introduces MambaAD, the first application of the Mamba framework to AD. MambaAD consists of a pre-trained encoder and a Mamba decoder, with a novel LSS module employed at different scales and depths. The LSS module, composed of sequential HSS modules and parallel multi-core convolutional networks, combines Mamba's global modeling prowess with CNN-based local feature correlation. The HSS module employs HS encoders to encode input features into five scanning patterns and eight directions, which facilitate the modeling of feature sequences in industrial products at their central positions. Extensive experiments on six diverse AD datasets and seven evaluation metrics demonstrate the effectiveness of our approach in achieving SoTA performance.

**Limitations, Broader Impact and Social Impact.** The model is not efficient enough and more lightweight models need to be designed. This study marks our initial attempt to apply Mamba in AD, laying a foundation for future research. We hope it can inspire lightweight designs in AD. MambaAD exhibits significant practical implications in enhancing industrial production efficiency.

## Acknowledgments and Disclosure of Funding

This work was supported by Jianbing Lingyan Foundation of Zhejiang Province, P.R. China (Grant No. 2023C01022).

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

# Appendix

## Overview

The supplementary material presents the following sections to strengthen the main manuscript:

- **Sec.** A shows more quantitative results for each category on the MVTec-AD dataset.
- **Sec.** B shows more quantitative results for each category on the VisA dataset.
- **Sec.** C shows more quantitative results for each category on the MVTec-3D dataset.
- **Sec.** D shows more quantitative results for each category on the Uni-Medical dataset.
- **Sec.** E shows more quantitative results for each category on the COCO-AD dataset.
- **Sec.** F shows more quantitative results for each category on the Real-IAD dataset.
- **Sec.** G shows more quantitative results for single-class results on the MVTec-AD dataset.
- **Sec.** H shows more quantitative results for single-class results on the VisA dataset.

## A    More Quantitative Results for Each Category on The MVTec-AD Dataset.

Tab. A1 and Tab. A2 respectively present the results of image-level anomaly detection and pixel-level anomaly localization quantitative outcomes across all categories within the MVTec-AD dataset. The results further demonstrate the superiority of our method over various SoTA approaches.

Table A1: Comparison with SoTA methods on **MVTec-AD** dataset for multi-class anomaly detection with AU-ROC/AP/F1_max metrics.

| | Method → | RD4AD [12] | UniAD [47] | SimpleNet [30] | DeSTSeg [55] | DiAD [18] | MambaAD |
| | Category ↓ | CVPR'22 | NeurIPS'22 | CVPR'23 | CVPR'23 | AAAI'24 | ours |
|---|---|---|---|---|---|---|---|
| Objects | Bottle | 99.6/99.9/98.4 | 99.7/100./100. | 100./100./100. | 98.7/99.6/96.8 | 99.7/96.5/91.8 | 100./100./100. |
| | Cable | 84.1/89.5/82.5 | 95.2/95.9/88.0 | 97.5/98.5/94.7 | 89.5/94.6/85.9 | 94.8/98.8/95.2 | 98.8/99.2/95.7 |
| | Capsule | 94.1/96.9/96.9 | 86.9/97.8/94.4 | 90.7/97.9/93.5 | 82.8/95.9/92.6 | 89.0/97.5/95.5 | 94.4/98.7/94.9 |
| | Hazelnut | 60.8/69.8/86.4 | 99.8/100./99.3 | 99.9/100./99.3 | 98.8/99.2/98.6 | 99.5/99.7/97.3 | 100./100./100. |
| | Metal Nut | 100./100./99.5 | 99.2/99.9/99.5 | 96.9/99.3/96.1 | 92.9/98.4/92.2 | 99.1/96.0/91.6 | 99.9/100./99.5 |
| | Pill | 97.5/99.6/96.8 | 93.7/98.7/95.7 | 88.2/97.7/92.5 | 77.1/94.4/91.7 | 95.7/98.5/94.5 | 97.0/99.5/96.2 |
| | Screw | 97.7/99.3/95.8 | 87.5/96.5/89.0 | 76.7/90.5/87.7 | 69.9/88.4/85.4 | 90.7/99.7/97.9 | 94.7/97.9/94.0 |
| | Toothbrush | 97.2/99.0/94.7 | 94.2/97.4/95.2 | 89.7/95.7/92.3 | 71.7/89.3/84.5 | 99.7/99.9/99.2 | 98.3/99.3/98.4 |
| | Transistor | 94.2/95.2/90.0 | 99.8/98.0/93.8 | 99.2/98.7/97.6 | 78.2/79.5/68.8 | 99.8/99.6/97.4 | 100./100./100. |
| | Zipper | 99.5/99.9/99.2 | 95.8/99.5/97.1 | 99.0/99.7/98.3 | 88.4/96.3/93.1 | 95.1/99.1/94.4 | 99.3/99.8/97.5 |
| Textures | Carpet | 98.5/99.6/97.2 | 99.8/99.9/99.4 | 95.7/98.7/93.2 | 95.9/98.3/94.9 | 99.4/99.9/98.3 | 99.8/99.9/99.4 |
| | Grid | 98.0/99.4/96.6 | 98.2/99.5/97.3 | 97.6/99.2/96.4 | 97.9/99.2/96.6 | 98.5/99.8/97.7 | 100./100./100. |
| | Leather | 100./100./100. | 100./100./100. | 100./100./100. | 99.2/99.8/98.9 | 99.8/99.7/97.6 | 100./100./100. |
| | Tile | 98.3/99.3/96.4 | 99.3/99.8/98.2 | 99.3/99.8/98.8 | 97.0/98.9/95.3 | 96.8/99.9/98.4 | 98.2/99.3/95.4 |
| | Wood | 99.2/99.8/98.3 | 98.6/99.6/96.6 | 98.4/99.5/96.7 | 99.9/100./99.2 | 99.7/100./100. | 98.8/99.6/96.6 |
| | Mean | 94.6/96.5/95.2 | 96.5/98.8/96.2 | 95.3/98.4/95.8 | 89.2/95.5/91.6 | 97.2/99.0/96.5 | 98.6/99.6/97.8 |

Table A2: Comparison with SoTA methods on **MVTec-AD** dataset for multi-class anomaly localization with AU-ROC/AP/F1_max/AU-PRO metrics.

| | Method → | RD4AD [12] | UniAD [47] | SimpleNet [30] | DeSTSeg [55] | DiAD [18] | MambaAD |
| | Category ↓ | CVPR'22 | NeurIPS'22 | CVPR'23 | CVPR'23 | AAAI'24 | (ours) |
|---|---|---|---|---|---|---|---|
| Objects | Bottle | 97.8/68.2/67.6/94.0 | 98.1/66.0/69.2/93.1 | 97.2/53.8/62.4/89.0 | 93.3/61.7/56.0/67.5 | 98.4/52.2/54.8/86.6 | 98.8/79.7/76.7/95.2 |
| | Cable | 85.1/26.3/33.6/75.1 | 97.3/39.9/45.2/86.1 | 96.7/42.4/51.2/85.4 | 89.3/37.5/40.5/49.4 | 96.8/50.1/57.8/80.5 | 95.8/42.2/48.1/90.3 |
| | Capsule | 98.8/43.4/50.1/94.8 | 98.5/42.7/46.5/92.1 | 98.5/35.4/44.3/84.5 | 95.8/47.9/48.9/62.1 | 97.1/42.0/45.3/87.2 | 98.4/43.9/47.7/92.6 |
| | Hazelnut | 97.9/36.2/51.6/92.7 | 98.1/55.2/56.8/94.1 | 98.4/44.6/51.4/87.4 | 98.2/65.8/61.6/84.5 | 98.3/79.2/80.4/91.5 | 99.0/63.6/64.4/95.7 |
| | Metal Nut | 93.8/62.3/65.4/91.9 | 94.8/55.5/66.4/81.8 | 98.0/83.1/79.4/85.2 | 84.2/42.0/22.8/53.0 | 97.3/30.0/38.3/90.6 | 96.7/74.5/79.1/93.7 |
| | Pill | 97.5/63.4/65.2/95.8 | 95.0/44.0/53.9/95.3 | 96.5/72.4/67.7/81.9 | 96.2/61.7/41.8/27.9 | 95.7/46.0/51.4/89.0 | 97.4/64.0/66.5/95.7 |
| | Screw | 99.4/40.2/44.7/96.8 | 98.3/28.7/37.6/95.2 | 96.5/15.9/23.2/84.0 | 93.8/19.9/25.3/47.3 | 97.9/60.6/59.6/95.0 | 99.5/49.8/50.9/97.1 |
| | Toothbrush | 99.0/53.6/58.8/92.0 | 98.4/34.9/45.7/87.9 | 98.4/46.9/52.5/87.4 | 96.2/52.9/58.8/30.9 | 99.0/78.7/72.8/95.0 | 99.0/48.5/59.2/91.7 |
| | Transistor | 85.9/42.3/45.2/74.7 | 97.9/59.5/64.6/93.5 | 95.8/58.2/56.0/83.2 | 73.6/38.4/39.2/43.9 | 95.1/15.6/31.7/90.0 | 96.5/69.4/67.1/87.0 |
| | Zipper | 98.5/53.9/60.3/94.1 | 96.8/40.1/49.9/92.6 | 97.9/53.4/54.6/90.7 | 97.3/64.7/59.2/66.9 | 96.2/60.7/60.0/91.6 | 98.4/60.4/61.7/94.3 |
| Textures | Carpet | 99.0/58.5/60.5/95.1 | 98.5/49.9/51.1/94.4 | 97.4/38.7/43.2/90.6 | 93.6/59.9/58.9/89.3 | 98.6/42.2/46.4/90.6 | 99.2/60.0/63.3/96.7 |
| | Grid | 99.2/46.0/47.4/97.0 | 96.5/23.0/28.4/92.9 | 96.8/20.5/27.6/88.6/ | 97.0/42.1/46.9/86.8 | 96.6/66.0/64.1/94.0 | 99.2/47.4/47.7/97.0 |
| | Leather | 99.3/38.0/45.1/97.4 | 98.8/32.9/34.4/96.8 | 98.7/28.5/32.9/92.7 | 99.5/71.5/66.5/91.1 | 98.8/56.1/62.3/91.3 | 99.4/50.3/53.3/98.7 |
| | Tile | 95.3/48.5/60.5/85.8 | 91.8/42.1/50.6/78.4 | 95.7/60.5/59.9/90.6 | 93.0/71.0/66.2/87.1 | 92.4/65.7/64.1/90.7 | 93.8/45.1/54.8/80.0 |
| | Wood | 95.3/47.8/51.0/90.0 | 93.2/37.2/41.5/86.7 | 91.4/34.8/39.7/76.3 | 95.9/77.3/71.3/83.4 | 93.3/43.3/43.5/97.5 | 94.4/46.2/48.2/91.2 |
| | Mean | 96.1/48.6/53.8/91.1 | 96.8/43.4/49.5/90.7 | 96.9/45.9/49.7/86.5 | 93.1/54.3/50.9/64.8 | 96.8/52.6/55.5/90.7 | 97.7/56.3/59.2/93.1 |

## B More Quantitative Results for Each Category on The VisA Dataset.

Tab. A3 and Tab. A4 respectively present the results of image-level anomaly detection and pixel-level anomaly localization quantitative outcomes across all categories within the VisA dataset. The results further demonstrate the superiority of our method over various SoTA approaches.

Table A3: Comparison with SoTA methods on **VisA** dataset for multi-class anomaly detection with AU-ROC/AP/F1_max metrics.

| Method → 
 Category ↓ | RD4AD [12] 
 CVPR'22 | UniAD [47] 
 NeurIPS'22 | SimpleNet [30] 
 CVPR'23 | DeSTSeg [55] 
 CVPR'23 | DiAD [18] 
 AAAI'24 | MambaAD 
 (ours) |
|---|---|---|---|---|---|---|
| pcb1 | 96.2/95.5/91.9 | 92.8/92.7/87.8 | 91.6/91.9/86.0 | 87.6/83.1/83.7 | 88.1/88.7/80.7 | 95.4/93.0/91.6 |
| pcb2 | **97.8/97.8/94.2** | 87.8/87.7/83.1 | 92.4/93.3/84.5 | 86.5/85.8/82.6 | 91.4/91.4/84.7 | 94.2/93.7/89.3 |
| pcb3 | **96.4/96.2/91.0** | 78.6/78.6/76.1 | 89.1/91.1/82.6 | 93.7/95.1/87.0 | 86.2/87.6/77.6 | 93.7/94.1/86.7 |
| pcb4 | **99.9/99.9/99.0** | 98.8/98.8/94.3 | 97.0/97.0/93.5 | 97.8/97.8/92.7 | 99.6/99.5/97.0 | **99.9/99.9**/98.5 |
| macaroni1 | 75.9/61.5/76.8 | 79.9/79.8/72.7 | 85.9/82.5/73.1 | 76.6/69.0/71.0 | 85.7/85.2/78.8 | **91.6/89.8/81.6** |
| macaroni2 | **88.3/84.5/83.8** | 71.6/71.6/69.9 | 68.3/54.3/59.7 | 68.9/62.1/67.7 | 62.5/57.4/69.6 | 81.6/78.0/73.8 |
| capsules | 82.2/90.4/81.3 | 55.6/55.6/76.9 | 74.1/82.8/74.6 | 87.1/93.0/84.2 | 58.2/69.0/78.5 | **91.8/95.0/88.8** |
| candle | 92.3/92.9/86.0 | 94.1/94.0/86.1 | 84.1/73.3/76.6 | 94.9/94.8/89.2 | 92.8/92.0/87.6 | **96.8/96.9/90.1** |
| cashew | 92.0/95.8/90.7 | 92.8/92.8/**91.4** | 88.0/91.3/84.7 | 92.0/96.1/88.1 | 91.5/95.7/89.7 | **94.5/97.3**/91.1 |
| chewinggum | 94.9/97.5/92.1 | 96.3/96.2/95.2 | 96.4/98.2/93.8 | 95.8/98.3/94.7 | **99.1/99.5/95.9** | 97.7/98.9/94.2 |
| fryum | 95.3/97.9/91.5 | 83.0/83.0/85.0 | 88.4/93.0/83.3 | 92.1/96.1/89.5 | 89.8/95.0/87.2 | 95.2/97.7/90.5 |
| pipe_fryum | 97.9/98.9/96.5 | 94.7/94.7/93.9 | 90.8/95.5/88.6 | 94.1/97.1/91.9 | 96.2/98.1/93.7 | **98.7/99.3/97.0** |
| Mean | 92.4/92.4/**89.6** | 85.5/85.5/84.4 | 87.2/87.0/81.8 | 88.9/89.0/85.2 | 86.8/88.3/85.1 | **94.3/94.5**/89.4 |

Table A4: Comparison with SoTA methods on **VisA** dataset for multi-class anomaly localization with AU-ROC/AP/F1_max/AU-PRO metrics.

| Method → 
 Category ↓ | RD4AD [12] 
 CVPR'22 | UniAD [47] 
 NeurIPS'22 | SimpleNet [30] 
 CVPR'23 | DeSTSeg [55] 
 CVPR'23 | DiAD [18] 
 AAAI'24 | MambaAD 
 (ours) |
|---|---|---|---|---|---|---|
| pcb1 | 99.4/66.2/62.4/**95.8** | 93.3/ 3.9/ 8.3/64.1 | 99.2/**86.1/78.8**/83.6 | 95.8/46.4/49.0/83.2 | 98.7/49.6/52.8/80.2 | **99.8**/77.1/72.4/92.8 |
| pcb2 | 98.0/**22.3/30.0**/90.8 | 93.9/ 4.2/ 9.2/66.9 | 96.6/ 8.9/18.6/85.7 | 97.3/14.6/28.2/79.9 | 95.2/ 7.5/16.7/67.0 | **98.9**/13.3/23.4/**89.6** |
| pcb3 | 97.9/26.2/35.2/93.9 | 97.3/13.8/21.9/70.6 | 97.2/**31.0/36.1**/85.1 | 97.7/28.1/33.4/62.4 | 96.7/ 8.0/18.8/68.9 | **99.1**/18.3/27.4/**89.1** |
| pcb4 | 97.8/31.4/37.0/**88.7** | 94.9/14.7/22.9/72.3 | 93.9/23.9/32.9/61.1 | 95.8/**53.0/53.2**/76.9 | 97.0/17.6/27.2/85.0 | 98.6/47.0/46.9/87.6 |
| macaroni1 | 99.4/ 2.9/ 6.9/95.3 | 97.4/ 3.7/ 9.7/84.0 | 98.9/ 3.5/ 8.4/92.0 | 99.1/ 5.8/13.4/62.4 | 94.1/10.2/16.7/68.5 | **99.5**/17.5/27.6/95.2 |
| macaroni2 | **99.7**/13.2/21.8/97.4 | 95.2/ 0.9/ 4.3/76.6 | 93.2/ 0.6/ 3.9/77.8 | 98.5/ 6.3/14.4/70.0 | 93.6/ 0.9/ 2.8/73.1 | 99.5/ 9.2/16.1/**96.2** |
| capsules | 99.4/60.4/60.8/93.1 | 88.7/ 3.0/ 7.4/43.7 | 97.1/52.9/53.3/73.7 | 96.9/33.2/39.1/76.7 | 97.3/10.0/21.0/77.9 | 99.1/**61.3**/59.8/91.8 |
| candle | **99.1**/25.3/35.8/94.9 | 98.5/17.6/27.9/91.6 | 97.6/ 8.4/16.5/87.6 | 98.7/**39.9/45.8**/69.0 | 97.3/12.8/22.8/89.4 | 99.0/23.2/32.4/**95.5** |
| cashew | 91.7/44.2/49.7/86.2 | 98.6/51.7/58.3/**87.9** | **98.9/68.9/66.0**/84.4 | 87.9/47.6/52.1/66.3 | 90.9/53.1/60.9/61.8 | 94.3/46.8/51.4/87.8 |
| chewinggum | 98.7/59.9/61.7/76.9 | **98.8**/54.9/56.1/**81.3** | 97.9/26.8/29.8/78.3 | **98.8/86.9/81.0**/68.3 | 94.7/11.9/25.8/59.5 | 98.1/57.5/59.9/79.7 |
| fryum | 97.0/47.6/51.5/93.4 | 95.9/34.0/40.6/76.2 | 93.0/39.1/45.4/85.1 | 88.1/35.2/38.5/47.7 | **97.6/58.6/60.1**/81.3 | 96.9/47.8/51.9/91.6 |
| pipe_fryum | 99.1/56.8/58.8/95.4 | 98.9/50.2/57.7/91.5 | 98.5/65.6/63.4/83.0 | 98.9/**78.8/72.7**/45.9 | **99.4**/72.7/69.9/89.9 | 99.1/53.5/58.5/95.1 |
| Mean | 98.1/38.0/42.6/**91.8** | 95.9/21.0/27.0/75.6 | 96.8/34.7/37.8/81.4 | 96.1/**39.6**/43.4/67.4 | 96.0/26.1/33.0/75.2 | **98.5**/39.4/**44.0**/91.0 |

## C More Quantitative Results for Each Category on The MVTec-3D Dataset.

Tab. A5 and Tab. A6 respectively present the results of image-level anomaly detection and pixel-level anomaly localization quantitative outcomes across all categories within the MVTec-3D dataset. The results further demonstrate the superiority of our method over various SoTA approaches.

Table A5: Comparison with SoTA methods on **MVTec-3D** dataset for multi-class anomaly detection with AU-ROC/AP/F1_max metrics.

| Method → 
 Category ↓ | RD4AD [12] 
 CVPR'22 | UniAD [47] 
 NeurIPS'22 | SimpleNet [30] 
 CVPR'23 | DeSTSeg [55] 
 CVPR'23 | DiAD [18] 
 AAAI'24 | MambaAD 
 (ours) |
|---|---|---|---|---|---|---|
| bagel | 82.5/95.4/89.6 | 82.7/95.8/89.3 | 76.2/93.3/89.3 | 89.7/97.4/92.4 | **100./100./100**. | 87.7/96.7/92.2 |
| cable gland | 90.1/97.5/92.6 | 89.8/97.2/**93.9** | 70.3/91.0/90.2 | 84.8/95.7/91.5 | 68.1/91.0/92.3 | **94.3/98.6**/93.5 |
| carrot | 87.3/96.7/93.2 | 76.8/93.8/92.5 | 71.4/92.6/91.2 | 79.1/94.7/91.0 | **94.4/99.3/98.0** | 90.7/97.7/95.0 |
| cookie | 46.0/77.4/88.0 | **77.3/93.5**/88.0 | 66.7/89.6/88.4 | 69.4/78.8/**90.9** | 69.4/78.8/**90.9** | 61.2/87.5/88.4 |
| dowel | 96.7/98.9/**97.6** | 96.7/99.3/96.2 | 83.7/95.1/91.7 | 77.3/94.3/88.9 | **98.0/99.3/97.3** | 97.6/99.5/96.6 |
| foam | 74.3/92.9/**90.6** | 70.5/92.4/88.9 | 77.4/94.2/89.7 | 77.9/94.7/88.9 | **100./100./100.** | 84.0/95.8/90.4 |
| peach | 64.3/84.8/90.6 | 70.0/91.0/90.5 | 62.0/86.9/89.7 | 82.2/95.3/90.7 | 58.0/91.3/**94.3** | **92.8/98.1/94.3** |
| potato | 62.5/88.5/90.5 | 51.6/81.8/89.3 | 56.7/82.2/89.8 | 62.9/87.8/90.9 | **76.3/94.3/95.0** | 66.8/88.6/90.5 |
| rope | 96.3/98.5/93.2 | **97.4/99.0/95.5** | 95.6/98.4/94.7 | 93.5/97.4/92.5 | 89.2/95.4/91.9 | 97.4/98.9/94.7 |
| tire | 79.2/93.0/88.4 | 75.7/90.7/89.7 | 65.3/86.8/87.9 | 80.8/93.6/91.5 | **92.7/98.9/95.8** | 90.0/97.0/91.9 |
| Mean | 77.9/92.4/91.4 | 78.9/93.4/91.4 | 72.5/91.0/90.3 | 79.6/94.1/90.6 | 84.6/94.8/**95.6** | **86.2/95.8**/92.8 |

Table A6: Comparison with SoTA methods on **MVTec-3D** dataset for multi-class anomaly localization with AU-ROC/AP/F1_max/AU-PRO metrics.

| Method → Category ↓ | RD4AD [12] CVPR'22 | UniAD [47] NeurIPS'22 | SimpleNet [30] CVPR'23 | DeSTSeg [55] CVPR'23 | DiAD [18] AAAI'24 | MambaAD (ours) |
|---|---|---|---|---|---|---|
| bagel | 98.6/39.0/45.1/91.3 | 97.6/30.4/35.8/84.4 | 93.2/23.6/30.9/70.4 | **98.7**/**53.0**/52.9/77.6 | 98.5/49.6/**54.2**/**93.8** | 98.5/38.3/41.1/92.1 |
| cable gland | **99.4**/37.9/43.2/98.2 | 98.9/26.4/34.8/96.3 | 95.2/14.4/23.1/86.8 | 97.8/**46.0**/**49.8**/64.1 | 98.4/25.2/32.0/94.5 | **99.4**/39.5/43.9/**98.4** |
| carrot | **99.4**/27.5/33.7/97.2 | 98.0/12.2/19.3/93.4 | 96.4/13.8/21.2/84.4 | 86.9/26.5/21.7/14.2 | 98.6/20.0/26.9/94.6 | **99.4**/30.1/35.4/**98.1** |
| cookie | 96.6/27.5/32.9/86.6 | **97.5**/**40.4**/**45.6**/88.7 | 90.5/26.7/31.4/66.6 | 93.3/34.0/35.6/40.9 | 94.3/14.0/23.8/83.5 | 96.8/39.0/41.9/83.6 |
| dowel | **99.7**/**47.7**/**50.8**/**98.8** | 99.1/32.1/37.7/96.1 | 95.3/17.4/25.6/83.0 | 97.3/43.1/44.5/31.2 | 97.2/31.4/40.1/89.6 | 99.6/49.9/50.4/97.1 |
| foam | 94.2/15.0/26.4/79.9 | 82.2/ 6.8/18.9/55.8 | 87.8/15.7/26.7/66.7 | **95.7**/**43.7**/**49.3**/63.6 | 89.8/ 9.6/23.5/69.1 | 95.1/23.4/32.8/82.7 |
| peach | 98.5/15.5/22.7/93.2 | 97.4/11.7/17.9/90.4 | 92.9/ 8.1/15.0/74.8 | 95.9/35.7/41.2/48.2 | 98.4/27.6/31.3/94.2 | **99.4**/**43.2**/**45.1**/**97.1** |
| potato | **99.1**/14.9/22.5/95.9 | 97.6/ 5.1/ 8.9/91.1 | 91.0/ 4.3/10.9/72.8 | 89.2/ 8.7/12.2/ 6.2 | 98.0/ 8.6/17.8/93.9 | 99.0/**17.6**/**22.6**/94.8 |
| rope | **99.6**/50.3/55.9/97.9 | 99.0/34.5/40.7/94.3 | 99.3/51.1/52.9/92.8 | 98.8/**64.5**/**62.1**/90.4 | 99.3/61.0/59.9/96.5 | 99.4/52.1/50.9/95.5 |
| tire | 99.2/23.2/31.1/96.4 | 98.0/11.9/20.3/90.6 | 93.8/ 8.1/15.3/77.9 | 97.0/25.8/30.0/27.3 | 91.8/ 5.9/13.7/68.8 | **99.5**/**42.0**/**46.9**/**97.0** |
| Mean | 98.4/29.8/36.4/93.5 | 96.5/21.2/28.0/88.1 | 93.6/18.3/25.3/77.6 | 95.1/**38.1**/39.9/46.4 | 96.4/25.3/32.3/87.8 | **98.6**/37.5/**41.1**/**93.6** |

# D More Quantitative Results for Each Category on The Uni-Medical Dataset.

Tab. A7 and Tab. A8 respectively present the results of image-level anomaly detection and pixel-level anomaly localization quantitative outcomes across all categories within the Uni-Medical dataset. The results further demonstrate the superiority of our method over various SoTA approaches.

Table A7: Comparison with SoTA methods on **Uni-Medical** dataset for multi-class anomaly detection with AU-ROC/AP/F1_max metrics.

| Method → Category ↓ | RD4AD [12] CVPR'22 | UniAD [47] NeurIPS'22 | SimpleNet [30] CVPR'23 | DeSTSeg [55] CVPR'23 | DiAD [18] AAAI'24 | MambaAD (ours) |
|---|---|---|---|---|---|---|
| brain | 82.4/94.4/91.5 | 89.9/97.5/92.6 | 82.3/95.6/90.9 | 84.5/95.0/92.1 | 93.7/98.1/95.0 | **94.2**/**98.6**/94.5 |
| liver | 55.1/46.3/64.1 | 61.0/48.8/63.2 | 55.8/47.6/60.9 | **69.2**/**60.6**/**64.7** | 59.2/55.6/60.9 | 63.2/53.1/**64.7** |
| retinal | 89.2/86.7/78.5 | 84.6/79.4/73.9 | 88.8/87.6/78.6 | 88.3/83.8/79.2 | 88.3/86.6/77.7 | **93.6**/**88.7**/86.6 |
| Mean | 75.6/75.8/78.0 | 78.5/75.2/76.6 | 75.6/76.9/76.8 | 80.7/79.8/78.7 | 80.4/**80.1**/77.8 | **83.7**/80.1/**82.0** |

Table A8: Comparison with SoTA methods on **Uni-Medical** dataset for multi-class anomaly localization with AU-ROC/AP/F1_max/AU-PRO metrics.

| Method → Category ↓ | RD4AD [12] CVPR'22 | UniAD [47] NeurIPS'22 | SimpleNet [30] CVPR'23 | DeSTSeg [55] CVPR'23 | DiAD [18] AAAI'24 | MambaAD (ours) |
|---|---|---|---|---|---|---|
| brain | 96.5/45.9/49.2/82.6 | 97.4/55.7/55.7/82.4 | 94.8/42.1/42.4/73.0 | 89.3/33.0/37.0/23.3 | 95.4/42.9/36.7/80.3 | **98.1**/**62.7**/**62.2**/**87.6** |
| liver | 96.6/ 5.7/10.3/89.9 | 97.1/ 7.8/13.7/**92.7** | **97.4**/13.2/20.1/86.3 | 79.4/**21.9**/**28.5**/20.3 | 97.1/13.7/ 7.3/91.4 | 96.9/ 9.1/16.3/91.8 |
| retinal | **96.4**/**64.7**/60.9/**86.5** | 94.8/49.3/51.3/79.9 | 95.5/59.5/56.3/82.1 | 91.0/59.0/46.8/31.7 | 95.3/57.5/62.8/84.1 | 95.7/64.5/**63.4**/83.1 |
| Mean | 96.5/38.7/40.1/86.4 | 96.4/37.6/40.2/85.0 | 95.9/38.3/39.6/80.5 | 86.6/38.0/37.5/25.1 | 95.9/38.0/35.6/85.4 | **96.9**/**45.4**/**47.3**/**87.5** |

# E More Quantitative Results for Each Category on The COCO-AD Dataset.

Tab. A9 and Tab. A10 respectively present the results of image-level anomaly detection and pixel-level anomaly localization quantitative outcomes across all categories within the COCO-AD dataset. The results further demonstrate the superiority of our method over various SoTA approaches.

Table A9: Comparison with SoTA methods on **COCO-AD** dataset for multi-class anomaly detection with AU-ROC/AP/F1_max metrics.

| Method → Category ↓ | RD4AD [12] CVPR'22 | UniAD [47] NeurIPS'22 | SimpleNet [30] CVPR'23 | DeSTSeg [55] CVPR'23 | DiAD [18] AAAI'24 | MambaAD (ours) |
|---|---|---|---|---|---|---|
| 0 | 65.7/81.9/85.1 | 66.1/84.0/85.1 | 57.8/77.4/84.7 | 59.7/79.1/85.0 | 57.5/77.5/**85.3** | **75.3**/**89.8**/85.2 |
| 1 | 54.9/46.8/61.1 | **56.1**/47.8/61.1 | 51.2/42.3/59.0 | 55.6/47.9/61.2 | 54.4/**49.8**/62.2 | 55.0/48.1/61.0 |
| 2 | 59.6/39.4/51.3 | 52.3/30.8/49.5 | 60.1/38.5/50.7 | 55.8/37.6/50.1 | 63.8/43.4/52.5 | **66.9**/**46.4**/**54.6** |
| 3 | 53.5/36.4/51.5 | 50.1/33.5/51.2 | 59.2/39.2/52.2 | 53.5/36.5/51.2 | **60.1**/41.4/52.9 | 58.4/40.5/51.9 |
| Mean | 58.4/51.1/62.3 | 56.2/49.0/61.7 | 57.1/49.4/61.7 | 56.2/50.3/61.9 | 58.9/53.0/63.2 | **63.9**/**56.2**/63.2 |

Table A10: Comparison with SoTA methods on **COCO-AD** dataset for multi-class anomaly localization with AU-ROC/AP/F1_max/AU-PRO metrics.

| Method → Category ↓ | RD4AD [12] CVPR'22 | UniAD [47] NeurIPS'22 | SimpleNet [30] CVPR'23 | DeSTSeg [55] CVPR'23 | DiAD [18] AAAI'24 | MambaAD (ours) |
|---|---|---|---|---|---|---|
| 0 | 72.1/30.8/38.2/45.9 | 70.8/29.4/36.7/36.5 | 64.0/27.4/34.4/26.8 | 61.8/21.5/27.7/23.5 | 67.0/33.4/26.2/28.8 | **75.6/38.6/41.7/46.2** |
| 1 | 70.7/ 6.2/11.2/40.6 | 70.0/ 6.2/11.3/31.8 | 61.4/ 4.9/ 8.9/33.0 | 69.3/ 6.8/11.3/27.7 | 71.3/11.8/ 7.8/28.8 | 71.2/ 6.6/11.3/36.6 |
| 2 | 68.4/11.6/18.9/42.9 | 60.9/ 7.7/14.7/27.0 | 57.4/ 8.2/14.4/29.2 | 61.1/ 9.8/13.9/26.3 | 68.0/**19.2**/12.2/33.2 | **71.2**/13.9/**21.6/44.4** |
| 3 | 58.3/ 8.4/14.2/33.4 | 59.8/ 8.3/**14.8**/31.4 | 55.3/ 8.2/13.9/21.0 | 51.2/ 6.9/12.4/16.8 | **65.9**/17.5/10.6/32.3 | 59.0/ 8.6/14.4/**34.7** |
| Mean | 67.4/14.3/20.6/**40.7** | 65.4/12.9/19.4/31.7 | 59.5/12.2/17.9/27.5 | 60.9/11.3/16.3/23.6 | 68.0/**20.5**/14.2/30.8 | **69.3**/16.9/**22.2**/40.5 |

# F  More Quantitative Results for Each Category on The Real-IAD Dataset.

Tab. A11 and Tab. A12 respectively present the results of image-level anomaly detection and pixel-level anomaly localization quantitative outcomes across all categories within the Real-IAD dataset. The results further demonstrate the superiority of our method over various SoTA approaches.

Table A11: Comparison with SoTA methods on **Real-IAD** dataset for multi-class anomaly detection with AU-ROC/AP/F1_max metrics.

| Method → Category ↓ | RD4AD [12] CVPR'22 | UniAD [47] NeurIPS'22 | SimpleNet [30] CVPR'23 | DeSTSeg [55] CVPR'23 | DiAD [18] AAAI'24 | MambaAD (ours) |
|---|---|---|---|---|---|---|
| audiojack | 76.2/63.2/60.8 | 81.4/**76.6**/64.9 | 58.4/44.2/50.9 | 81.1/72.6/64.5 | 76.5/54.3/65.7 | **84.2**/76.5/**67.4** |
| bottle cap | 89.5/86.3/81.0 | 92.5/91.7/81.7 | 54.1/47.6/60.3 | 78.1/74.6/68.1 | 91.6/**94.0/87.9** | **92.8**/92.0/82.1 |
| button battery | 73.3/78.9/76.1 | 75.9/81.6/76.3 | 52.5/60.5/72.4 | **86.7/89.2/83.5** | 80.5/71.3/70.6 | 79.8/85.3/77.8 |
| end cap | 79.8/84.0/77.8 | 80.9/**86.1**/78.0 | 51.6/60.8/72.9 | 77.9/81.1/77.1 | **85.1**/83.4/**84.8** | 78.0/82.8/77.2 |
| eraser | 90.0/88.7/79.7 | 90.3/89.2/80.2 | 46.4/39.1/55.8 | 84.6/82.9/71.8 | 80.0/80.0/77.3 | 87.5/86.2/76.1 |
| fire hood | 78.3/70.1/64.5 | 80.6/74.8/66.4 | 58.1/41.9/54.4 | 81.7/72.4/67.7 | **83.3/81.7/80.5** | 79.3/72.5/64.8 |
| mint | 65.8/63.1/64.8 | 67.0/66.6/64.6 | 52.4/50.3/63.7 | 58.4/55.8/63.7 | **76.7/76.7/76.0** | 70.1/70.8/65.5 |
| mounts | **88.6/79.9**/74.8 | 87.6/77.3/77.2 | 58.7/48.1/52.4 | 74.7/56.5/63.1 | 75.3/74.5/**82.5** | 86.8/78.0/73.5 |
| pcb | 79.5/85.8/79.7 | 81.0/88.2/79.1 | 54.5/66.0/75.5 | 82.0/82.7/79.6 | 86.0/85.1/**85.4** | **89.1/93.7/**84.0 |
| phone battery | 87.5/83.3/77.1 | 83.6/80.0/71.6 | 51.6/43.8/58.0 | 83.3/81.8/72.1 | 82.3/77.7/75.9 | **90.2/88.9/80.5** |
| plastic nut | 80.3/68.0/64.4 | 80.0/69.2/63.7 | 59.2/40.3/51.8 | 83.1/75.4/66.5 | 71.9/58.2/65.6 | **87.1/80.7/70.7** |
| plastic plug | 81.9/74.3/68.8 | 81.4/75.9/67.6 | 48.2/38.4/54.6 | 71.7/63.1/60.0 | **88.7/89.2/90.9** | 85.7/82.2/72.6 |
| porcelain doll | 86.3/76.3/71.5 | 85.1/75.2/69.3 | 66.3/54.5/52.1 | 78.7/66.2/64.3 | 72.6/66.8/65.2 | **88.0/82.2/74.1** |
| regulator | 66.9/48.8/47.7 | 56.9/41.5/44.5 | 50.5/29.0/43.9 | **79.2**/63.5/56.9 | 72.1/71.4/**78.2** | 69.7/58.7/50.4 |
| rolled strip base | 97.5/98.7/94.7 | **98.7/99.3/96.5** | 59.0/75.7/79.8 | 96.5/96.2/93.0 | 68.4/55.9/56.8 | 98.0/99.0/95.0 |
| sim card set | 91.6/91.8/84.8 | 89.7/90.3/83.2 | 63.1/69.7/70.8 | **95.5/96.2/89.2** | 72.6/53.7/61.5 | 94.4/95.1/87.2 |
| switch | 84.3/87.2/77.9 | 85.5/88.6/78.4 | 62.2/66.8/68.6 | 90.1/92.8/83.1 | 73.4/49.4/61.2 | **91.7/94.0/85.4** |
| tape | 96.0/95.1/87.6 | **97.2/96.2/89.4** | 49.9/41.1/54.5 | 94.5/93.4/85.9 | 73.9/57.8/66.1 | 96.8/95.9/89.3 |
| terminalblock | 89.4/89.7/83.1 | 87.5/89.1/81.0 | 59.8/64.7/68.8 | 83.1/86.2/76.6 | 62.1/36.4/47.8 | **96.1/96.8/90.0** |
| toothbrush | 82.0/83.8/77.2 | 78.4/80.1/75.6 | 65.9/70.0/70.1 | 83.7/85.3/79.0 | **91.2/93.7/90.9** | 85.1/86.2/80.3 |
| toy | 69.4/74.2/75.9 | 68.4/75.1/74.8 | 57.8/64.4/73.4 | 70.3/74.8/75.4 | 66.2/57.3/59.8 | **83.0/87.5/79.6** |
| toy brick | 63.6/56.1/59.0 | **77.0/71.1/66.2** | 58.3/49.7/58.2 | 73.2/68.7/63.3 | 68.4/45.3/55.9 | 70.5/63.7/61.6 |
| transistor1 | 91.0/94.0/85.1 | 93.7/95.9/88.9 | 62.2/69.2/72.1 | 90.2/92.1/84.6 | 73.1/63.1/62.7 | **94.4/96.0/89.0** |
| u block | 89.5/85.0/74.2 | 88.8/84.2/**75.5** | 62.4/48.4/51.8 | 80.1/73.9/64.3 | 75.2/68.4/67.9 | **89.7**/85.7/75.3 |
| usb | 84.9/84.3/75.1 | 78.7/79.4/69.1 | 57.0/55.3/62.9 | 87.8/88.0/78.3 | 58.9/37.4/45.7 | **92.0/92.2/84.5** |
| usb adaptor | 71.1/61.4/62.2 | 76.8/71.3/64.9 | 47.5/38.4/56.5 | **80.1**/74.9/**67.4** | 76.9/60.2/67.2 | 79.4/**76.0**/66.3 |
| vcpill | 85.1/80.3/72.4 | 87.1/84.0/74.7 | 59.0/48.7/56.4 | 83.8/81.5/69.9 | 64.1/40.4/56.2 | **88.3/87.7/77.4** |
| wooden beads | 81.2/78.9/70.9 | 78.4/77.2/67.8 | 55.1/52.0/60.2 | 82.4/78.5/73.0 | 62.1/56.4/65.9 | **82.5/81.7/**71.8 |
| woodstick | 76.9/61.2/58.1 | **80.8/72.6/63.6** | 58.2/35.6/45.2 | 80.4/69.2/60.3 | 74.1/66.0/62.1 | 80.4/69.0/63.4 |
| zipper | 95.3/97.2/91.2 | 98.2/98.9/95.3 | 77.2/86.7/77.6 | 96.9/98.1/93.5 | 86.0/87.0/84.0 | **99.2/99.6/96.9** |
| Mean | 82.4/79.0/73.9 | 83.0/80.9/74.3 | 57.2/53.4/61.5 | 82.3/79.2/73.2 | 75.6/66.4/69.9 | **86.3/84.6/77.0** |

# G  More Quantitative Results for Each Category on The MVTec-AD Dataset for Single-class Anomaly Detection.

Tab. A13 and Tab. A14 respectively present the results of image-level anomaly detection and pixel-level anomaly localization quantitative outcomes for single-class anomaly detection of the MVTec-AD dataset.

Table A12: Comparison with SoTA methods on **Real-IAD** dataset for multi-class anomaly localization with AU-ROC/AP/F1_max/AU-PRO metrics.

| Method → | RD4AD [12] | UniAD [47] | SimpleNet [30] | DeSTSeg [55] | DiAD [18] | MambaAD |
|---|---|---|---|---|---|---|
| Category ↓ | CVPR'22 | NeurIPS'22 | CVPR'23 | CVPR'23 | AAAI'24 | (ours) |
| audiojack | 96.6/12.8/22.1/79.6 | 97.6/20.0/31.0/83.7 | 74.4/0.9/4.8/38.0 | 95.5/25.4/31.9/52.6 | 91.6/1.0/3.9/63.3 | 97.7/21.6/29.5/83.9 |
| bottle cap | 99.5/18.9/29.9/95.7 | 99.5/19.4/29.6/96.0 | 85.3/2.3/5.7/45.1 | 94.5/25.3/31.1/25.3 | 94.6/4.9/11.4/73.0 | 99.7/30.6/34.6/97.2 |
| button battery | 97.6/33.8/37.8/86.5 | 76.7/28.5/34.4/77.5 | 75.9/3.2/6.6/40.5 | 98.3/63.9/60.4/36.9 | 84.1/1.4/5.3/66.9 | 98.1/46.7/49.5/86.2 |
| end cap | 96.7/12.5/22.5/89.2 | 95.8/8.8/17.4/85.4 | 63.1/0.5/2.8/25.7 | 89.6/14.4/22.7/29.5 | 81.3/2.0/6.9/38.2 | 97.0/12.0/19.6/89.4 |
| eraser | 99.5/30.8/36.7/96.0 | 99.3/24.4/30.9/94.1 | 80.6/2.7/7.1/42.8 | 95.8/52.7/53.9/46.7 | 91.1/7.7/15.4/67.5 | 99.2/30.2/38.3/93.7 |
| fire hood | 98.9/27.7/35.2/87.9 | 98.6/23.4/32.2/85.3 | 70.5/0.3/2.2/25.3 | 97.3/27.1/35.3/34.7 | 91.8/3.2/9.2/66.7 | 98.7/25.1/31.3/86.3 |
| mint | 95.0/11.7/23.0/72.3 | 94.4/7.7/18.1/62.3 | 79.9/0.9/3.6/43.3 | 84.1/10.3/22.4/9.9 | 91.1/5.7/11.6/64.2 | 96.5/15.9/27.0/72.6 |
| mounts | 99.3/30.6/37.1/94.9 | 99.4/28.0/32.8/95.2 | 80.5/2.2/6.8/46.1 | 94.2/30.0/41.3/43.3 | 84.3/0.4/1.1/48.8 | 99.2/31.4/35.4/93.5 |
| pcb | 97.5/15.8/24.3/88.3 | 97.0/18.5/28.1/81.6 | 78.0/1.4/4.3/41.3 | 97.2/37.1/40.4/48.8 | 92.0/3.7/7.4/66.5 | 99.2/46.3/50.4/93.1 |
| phone battery | 77.3/22.6/31.7/94.5 | 85.5/11.2/21.6/88.5 | 43.4/0.1/0.9/11.8 | 79.5/25.6/33.8/39.5 | 96.8/5.3/11.4/85.4 | 99.4/36.3/41.3/95.3 |
| plastic nut | 98.8/21.1/29.6/91.0 | 98.4/20.6/27.1/88.9 | 77.4/0.6/3.6/41.5 | 96.5/44.8/45.7/38.4 | 81.1/0.4/3.4/38.6 | 99.4/33.1/37.3/96.1 |
| plastic plug | 99.1/20.5/28.4/94.9 | 98.6/17.4/26.1/90.3 | 78.6/0.7/1.9/38.8 | 91.9/20.1/27.3/21.0 | 92.9/8.7/15.0/66.1 | 99.0/24.2/31.7/91.5 |
| porcelain doll | 99.2/24.8/34.6/95.7 | 98.7/14.1/24.5/93.2 | 81.8/2.0/6.4/47.0 | 93.1/35.9/40.3/24.8 | 93.1/1.4/4.8/70.4 | 99.2/31.3/36.6/95.4 |
| regulator | 98.0/7.8/16.1/88.6 | 95.5/9.1/17.4/76.1 | 76.6/0.1/0.6/38.1 | 88.8/18.9/23.6/17.5 | 84.2/0.4/1.5/44.4 | 97.6/20.6/29.8/87.0 |
| rolled strip base | 99.7/31.4/39.9/98.4 | 99.6/20.7/32.2/97.8 | 80.5/1.7/5.1/52.1 | 99.2/48.7/50.1/55.5 | 87.7/0.6/3.2/63.4 | 99.7/37.4/42.5/98.8 |
| sim card set | 98.5/40.2/44.2/89.5 | 97.9/31.6/39.8/85.0 | 71.0/6.8/14.3/30.8 | 99.1/65.5/62.1/73.9 | 89.9/1.7/5.8/60.4 | 98.8/51.1/50.6/89.4 |
| switch | 94.4/18.9/26.6/90.9 | 98.1/33.8/40.6/90.7 | 71.7/3.7/9.3/44.2 | 97.4/57.6/56.6/44.7 | 90.5/1.4/5.3/64.2 | 98.2/39.9/45.4/92.9 |
| tape | 99.7/42.4/47.8/98.4 | 99.7/29.2/36.9/97.5 | 77.5/1.2/3.9/41.4 | 99.0/61.7/57.6/48.2 | 81.7/0.4/2.7/47.3 | 99.8/47.1/48.2/98.0 |
| terminalblock | 99.5/27.4/35.8/97.6 | 99.2/23.1/30.5/94.4 | 87.0/0.8/3.6/54.8 | 96.6/40.6/44.1/34.8 | 75.5/0.1/1.1/38.5 | 99.8/35.3/39.7/98.2 |
| toothbrush | 96.9/26.1/34.2/88.7 | 95.7/16.4/25.3/84.3 | 84.7/7.2/14.8/52.6 | 94.3/30.0/37.3/42.8 | 82.0/1.9/6.6/54.5 | 97.5/27.8/36.7/91.4 |
| toy | 95.2/5.1/12.8/82.3 | 93.4/4.6/12.4/70.5 | 67.7/0.1/0.4/25.0 | 86.3/8.1/15.9/16.4 | 82.1/1.1/4.2/50.3 | 96.0/16.4/25.8/86.3 |
| toy brick | 96.4/16.0/24.6/75.3 | 97.4/17.1/27.6/81.3 | 86.5/5.2/11.1/56.3 | 94.7/24.6/30.8/45.5 | 93.5/3.1/8.1/66.4 | 96.6/18.0/25.8/74.7 |
| transistor1 | 99.1/29.6/35.5/95.1 | 98.9/25.6/33.2/94.3 | 71.7/5.1/11.3/35.3 | 97.3/43.8/44.5/45.4 | 88.6/7.2/15.3/58.1 | 99.4/39.4/40.0/96.5 |
| u block | 99.6/40.5/45.2/96.9 | 99.3/22.3/29.6/94.3 | 76.2/4.8/12.2/34.0 | 96.9/57.1/55.7/38.5 | 88.8/1.6/5.4/54.2 | 99.5/37.8/46.1/95.4 |
| usb | 98.1/26.4/35.2/91.0 | 97.9/20.6/31.7/85.3 | 81.1/1.5/4.9/52.4 | 98.4/42.2/47.7/57.1 | 78.0/1.0/3.1/28.0 | 99.2/39.1/44.4/95.2 |
| usb adaptor | 94.5/9.8/17.9/73.1 | 96.6/10.5/19.0/78.4 | 67.9/0.2/1.3/28.9 | 94.9/25.5/34.9/36.4 | 86.2/2.3/6.6/75.5 | 97.3/15.3/22.6/82.5 |
| vcpill | 98.3/43.1/48.6/88.7 | 99.1/40.7/43.0/91.3 | 68.2/1.1/3.3/22.0 | 97.1/64.7/62.3/42.3 | 90.2/1.3/5.2/60.8 | 98.7/50.2/54.5/89.3 |
| wooden beads | 98.0/27.1/34.7/85.7 | 97.6/16.5/23.6/84.6 | 68.1/2.4/6.0/28.3 | 94.7/38.9/42.9/39.4 | 85.0/1.1/4.7/45.6 | 98.0/32.6/39.8/84.5 |
| woodstick | 97.8/30.7/38.4/85.0 | 94.0/36.2/44.3/77.2 | 76.1/1.4/6.0/32.0 | 97.9/60.3/60.0/51.0 | 90.9/2.6/8.0/60.7 | 97.7/40.1/44.9/82.7 |
| zipper | 99.1/44.7/50.2/96.3 | 98.4/32.5/36.1/95.1 | 89.9/23.3/31.2/55.5 | 98.2/35.3/39.0/78.5 | 90.2/12.5/18.8/53.5 | 99.3/58.2/61.3/97.6 |
| mean | 97.3/25.0/32.7/89.6 | 97.3/21.1/29.2/86.7 | 75.7/2.8/6.5/39.0 | 94.6/37.9/41.7/40.6 | 88.0/2.9/7.1/58.1 | 98.5/33.0/38.7/90.5 |

Table A13: Comparison with SoTA methods on **MVTec-AD** dataset for **single-class** anomaly detection with AU-ROC/AP/F1_max metrics.

| | Method → | RD4AD [12] | UniAD [47] | SimpleNet [30] | DeSTSeg [55] | PatchCore [36] | MambaAD |
|---|---|---|---|---|---|---|---|
| | Category ↓ | CVPR'22 | NeurIPS'22 | CVPR'23 | CVPR'23 | CVPR'22 | ours |
| Objects | Bottle | 100./100./100. | 100./100./100. | 99.9/100./99.2 | 100./100./100. | 100./100./100. | 100./100./100. |
| | Cable | 94.8/97.0/89.7 | 97.9/98.7/93.8 | 99.6/99.7/97.3 | 97.1/98.4/93.3 | 99.8/99.9/98.9 | 98.5/99.1/95.1 |
| | Capsule | 98.4/99.7/97.3 | 87.1/95.3/95.2 | 97.2/99.2/98.6 | 97.4/99.5/96.8 | 97.4/99.4/98.2 | 97.8/99.5/97.2 |
| | Hazelnut | 100./100./100. | 99.7/99.8/98.6 | 99.0/99.4/97.2 | 99.7/99.8/99.3 | 100./100./100. | 100./100./100. |
| | Metal Nut | 99.9/100./99.5 | 98.7/99.7/98.4 | 99.9/100./99.5 | 99.7/99.9/98.9 | 99.9/100./99.5 | 100./100./99.5 |
| | Pill | 97.5/99.6/97.1 | 96.0/99.3/95.8 | 98.6/99.7/97.9 | 96.2/99.3/95.7 | 96.6/99.4/96.5 | 95.3/99.1/95.3 |
| | Screw | 97.7/99.2/95.9 | 87.7/93.0/93.1 | 97.5/99.2/95.1 | 92.3/97.5/90.9 | 98.1/99.3/97.5 | 97.6/99.1/97.1 |
| | Toothbrush | 93.3/97.2/95.2 | 88.1/94.8/93.8 | 99.4/99.8/98.4 | 99.7/99.9/98.4 | 100./100./100. | 96.1/98.4/95.2 |
| | Transistor | 98.4/97.7/93.8 | 100./100./100. | 99.9/99.9/98.7 | 98.5/98.2/94.7 | 100./100./100. | 100./100./100. |
| | Zipper | 98.3/99.5/97.9 | 90.6/96.5/93.9 | 99.8/99.9/99.6 | 100./100./100. | 99.6/99.9/98.7 | 98.6/99.6/97.5 |
| Textures | Carpet | 99.7/99.9/98.3 | 99.4/99.8/98.9 | 99.1/99.7/98.3 | 99.3/99.8/97.7 | 98.5/99.6/97.1 | 100./100./100. |
| | Grid | 100./100./100. | 97.6/99.3/96.6 | 100./100./100. | 100./100./100. | 99.7/99.9/99.1 | 100./100./100. |
| | Leather | 100./100./100. | 100./100./100. | 100./100./100. | 100./100./100. | 100./100./100. | 100./100./100. |
| | Tile | 100./100./100. | 95.8/98.2/95.8 | 99.9/99.9/98.8 | 100./100./99.4 | 99.0/99.7/98.8 | 97.5/99.1/94.7 |
| | Wood | 99.5/99.8/98.3 | 97.9/99.4/95.9 | 99.9/100./99.2 | 99.3/99.8/98.4 | 99.1/99.7/97.5 | 99.6/99.9/99.2 |
| | Mean | 98.5/99.3/97.5 | 95.8/98.9/96.7 | 99.3/99.8/98.5 | 98.6/99.5/97.6 | 99.2/99.8/98.8 | 98.7/99.6/98.1 |

Table A14: Comparison with SoTA methods on **MVTec-AD** dataset for **single-class** anomaly localization with AU-ROC/AP/F1_max/AU-PRO metrics.

| | Method → | RD4AD [12] | UniAD [47] | SimpleNet [30] | DeSTSeg [55] | PatchCore [36] | MambaAD |
|---|---|---|---|---|---|---|---|
| | Category ↓ | CVPR'22 | NeurIPS'22 | CVPR'23 | CVPR'23 | CVPR'22 | (ours) |
| Objects | Bottle | 98.6/75.9/74.1/95.8 | 98.3/73.6/70.7/94.4 | 98.0/70.4/72.7/88.8 | 99.3/91.0/84.3/94.0 | 98.5/77.7/75.2/94.9 | 98.8/79.7/76.6/96.1 |
| | Cable | 96.8/51.3/57.3/88.8 | 97.4/54.8/56.9/87.7 | 97.4/66.8/60.0/87.7 | 95.8/55.0/52.8/82.4 | 98.4/66.3/65.1/92.5 | 98.0/54.4/61.7/93.4 |
| | Capsule | 98.9/48.1/49.9/95.3 | 98.0/35.2/40.1/86 | 98.9/42.5/49.1/92.8 | 98.8/52.4/57.9/71.6 | 99.0/44.7/50.9/91.7 | 98.6/45.0/47.5/94.3 |
| | Hazelnut | 98.8/59.6/60.8/96.5 | 98.3/55.0/56.1/92.8 | 97.9/46.2/50.3/78.9 | 99.4/84.5/79.0/90.6 | 98.7/53.5/59.2/89.7 | 99.0/61.8/63.9/95.8 |
| | Metal Nut | 96.9/75.3/79.5/94.1 | 94.4/55.8/68.8/77.8 | 98.8/91.6/86.6/84.0 | 99.2/95.0/88.7/95.0 | 98.3/86.9/85.1/91.4 | 97.1/78.3/80.6/93.6 |
| | Pill | 97.6/66.6/66.7/95.9 | 95.2/45.9/51.9/92.2 | 98.5/79.5/72.6/92.5 | 98.9/87.9/79.6/56.6 | 97.8/77.8/72.9/94.1 | 97.3/64.5/66.0/95.8 |
| | Screw | 99.5/44.6/47.2/96.9 | 98.4/15.5/23.6/91.3 | 99.3/35.0/38.8/95.2 | 97.3/54.2/53.3/53.6 | 99.5/36.5/40.1/96.6 | 99.5/52.2/51.7/97.6 |
| | Toothbrush | 99.0/51.8/60.9/92.0 | 98.5/42.6/52.1/84.6 | 98.5/41.7/52.9/92.4 | 99.5/78.2/73.7/90.8 | 98.6/38.3/52.3/92.3 | 99.0/51.1/60.7/92.3 |
| | Transistor | 90.6/52.8/54.9/80.8 | 98.8/79.5/75.8/94.5 | 96.8/67.4/62.1/91.1 | 88.1/63.6/60.0/80.9 | 96.2/66.4/61.5/89.8 | 96.8/69.7/67.5/91.6 |
| | Zipper | 98.7/53.0/59.0/94.6 | 96.5/35.5/43.4/89.3 | 98.9/64.3/65.4/95.5 | 99.2/85.7/77.4/84.3 | 98.9/62.9/66.1/94.3 | 98.1/55.6/58.8/94.8 |
| Textures | Carpet | 99.1/60.1/60.9/95.6 | 98.5/52.1/53.3/94.9 | 98.0/43.2/47.8/87.9 | 98.2/77.5/72.5/94.9 | 99.1/64.1/63.0/94.8 | 99.2/63.0/63.4/97.0 |
| | Grid | 99.3/45.2/47.7/96.3 | 94.3/24.3/31.5/85.8 | 98.8/34.7/39.3/93.9 | 99.5/62.5/63.5/93.8 | 98.8/31.0/35.3/94.2 | 99.2/48.9/48.8/97.3 |
| | Leather | 99.3/44.7/46.7/97.9 | 99.1/38.5/41.9/98.1 | 99.2/41.7/45.6/95.7 | 99.8/74.9/69.6/96.9 | 99.3/46.3/46.7/96.8 | 99.3/47.5/49.5/98.6 |
| | Tile | 95.1/48.6/59.7/85.1 | 90.2/40.4/48.7/77.1 | 97.2/69.8/68.8/89.7 | 99.4/94.6/87.1/89.1 | 95.8/55.0/64.7/90.0 | 93.1/42.8/52.3/80.3 |
| | Wood | 94.8/48.6/49.5/91.6 | 93.5/38.6/43.9/85.2 | 94.0/45.6/48.1/82.5 | 98.4/84.5/77.5/90.9 | 95.0/50.2/50.9/85.3 | 93.9/44.2/47.7/92.1 |
| | Mean | 97.5/55.1/58.3/93.1 | 96.6/45.8/50.6/88.8 | 98.0/56.0/57.3/89.9 | 98.1/76.1/71.8/84.4 | 98.1/57.2/59.3/92.6 | 97.8/57.2/59.8/94.0 |

# H More Quantitative Results for Each Category on The VisA Dataset for Single-class Anomaly Detection.

Tab. A15 and Tab. A16 respectively present the results of image-level anomaly detection and pixel-level anomaly localization quantitative outcomes for single-class anomaly detection of the VisA dataset.

Table A15: Comparison with SoTA methods on **VisA** dataset for **single-class** anomaly detection with AU-ROC/AP/F1_max metrics.

| Method → Category ↓ | RD4AD [12] CVPR'22 | UniAD [47] NeurIPS'22 | SimpleNet [30] CVPR'23 | DeSTSeg [55] CVPR'23 | PatchCore [36] CVPR'22 | MambaAD (ours) |
|---|---|---|---|---|---|---|
| pcb1 | 95.9/95.6/92.8 | 91.9/90.9/87.2 | **98.9/98.9/96.0** | 94.9/92.5/92.0 | 97.7/97.1/94.9 | 96.0/95.1/93.8 |
| pcb2 | 96.5/96.2/92.1 | 92.2/92.3/84.8 | **98.8/98.8/93.9** | 96.6/94.7/93.4 | 97.7/98.2/93.3 | 97.1/96.8/91.9 |
| pcb3 | 96.2/96.1/91.1 | 90.9/91.3/86.2 | **98.9/98.8/95.1** | 97.0/97.3/91.7 | 98.5/98.5/93.5 | 97.8/98.0/92.7 |
| pcb4 | **100./100./99.5** | 98.8/98.6/96.1 | 99.3/99.0/97.6 | 99.3/99.3/94.8 | 99.8/99.8/97.5 | 99.7/99.7/97.5 |
| macaroni1 | 92.1/90.2/85.4 | 87.5/83.3/79.6 | **94.8/92.7/89.0** | 82.8/75.1/77.2 | 91.3/85.9/87.0 | 91.2/88.4/84.1 |
| macaroni2 | **86.1**/80.3/**80.2** | 83.1/**82.5**/76.4 | 83.4/77.0/70.4 | 72.8/66.1/69.2 | 75.2/60.6/59.4 | 81.8/78.9/77.7 |
| capsules | 89.7/94.2/87.7 | 79.1/88.7/79.0 | 90.1/93.3/86.4 | 85.3/91.5/83.6 | 77.7/83.2/74.4 | **91.4/94.6/90.5** |
| candle | 94.6/94.7/89.4 | 96.5/96.5/89.3 | 92.3/85.6/86.9 | 94.6/95.1/87.2 | 92.2/85.3/84.4 | **97.7/97.7/92.1** |
| cashew | **96.9/98.6**/93.4 | 91.9/96.1/90.3 | 94.1/96.2/90.3 | 73.8/87.2/81.3 | 91.2/93.5/89.2 | 95.7/97.6/**93.7** |
| chewinggum | 98.8/99.4/96.3 | 97.4/98.8/94.8 | 98.6/99.3/**96.9** | 97.1/98.8/95.9 | 97.9/98.8/95.4 | **99.0/99.5**/96.1 |
| fryum | 94.8/97.7/91.7 | 85.5/92.6/86.0 | 96.4/98.0/**93.5** | 91.7/96.0/88.6 | 95.0/96.7/89.8 | **96.5/98.3**/92.5 |
| pipe_fryum | 99.7/99.8/98.5 | 91.9/95.4/90.8 | **99.8/99.9/99.0** | 99.3/99.6/97.5 | 99.1/99.5/97.5 | 99.2/99.6/97.6 |
| Mean | 95.1/95.2/91.5 | 90.6/92.3/86.7 | **95.4**/94.8/91.3 | 90.4/91.1/87.7 | 92.8/91.4/88.0 | 95.3/**95.3/91.7** |

Table A16: Comparison with SoTA methods on **VisA** dataset for **single-class** anomaly localization with AU-ROC/AP/F1_max/AU-PRO metrics.

| Method → Category ↓ | RD4AD [12] CVPR'22 | UniAD [47] NeurIPS'22 | SimpleNet [30] CVPR'23 | DeSTSeg [55] CVPR'23 | PatchCore [36] CVPR'22 | MambaAD (ours) |
|---|---|---|---|---|---|---|
| pcb1 | 99.7/71.2/68.4/94.8 | 99.2/53.0/56.2/86.7 | 99.6/87.7/78.4/89.6 | 99.4/61.5/62.8/76.9 | **99.8/**91.8/85.9/94.7 | **99.8/**72.5/67.4/**94.9** |
| pcb2 | 98.8/16.2/24.0/90.1 | 97.1/ 5.8/13.0/80.8 | 98.2/11.9/22.2/87.7 | 98.9/**20.0/30.1/91.5** | 98.7/12.5/22.8/89.9 | **98.9**/10.1/19.5/90.7 |
| pcb3 | 99.2/26.4/29.6/92.0 | 98.2/14.9/23.0/80.3 | 99.3/45.5/45.4/92.5 | 98.7/31.3/33.1/40.4 | **99.4/48.1/47.3/**89.2 | 99.3/25.7/27.9/**92.8** |
| pcb4 | 98.4/39.7/40.9/89.1 | 97.5/20.8/29.4/82.0 | 97.5/40.6/45.5/80.6 | 96.7/27.5/37.5/66.2 | 98.2/42.1/45.8/82.2 | **98.7/47.6/47.0/90.9** |
| macaroni1 | 99.6/**21.0/31.1**/95.4 | 99.0/ 6.8/13.5/95.8 | 99.6/ 7.0/11.1/98.5 | 98.7/17.3/25.3/53.4 | 99.7/ 7.8/13.5/98.4 | 99.6/18.7/27.8/**96.8** |
| macaroni2 | **99.5/**10.9/18.7/**95.9** | 98.2/ 3.8/10.3/95.3 | 98.4/ 3.9/ 8.8/94.8 | 99.0/ 8.6/18.2/68.8 | 98.8/ 4.9/ 7.6/94.5 | 99.3/10.8/16.2/95.9 |
| capsules | **99.6**/61.1/59.5/92.8 | 98.4/44.8/49.4/76.6 | 98.8/52.8/56.2/89.6 | 98.9/49.6/51.0/90.0 | 99.4/**65.2/65.6**/89.8 | 99.3/**65.2/**62.5/**93.1** |
| candle | 98.9/22.6/33.7/94.5 | 98.9/16.7/26.2/94.9 | 98.4/13.1/22.2/90.2 | 99.0/**31.1/36.7/**78.7 | **99.3/**18.7/27.2/96.7 | 98.7/19.9/30.5/**95.1** |
| cashew | 95.4/53.0/56.2/89.8 | 99.2/64.6/64.3/85.5 | 98.9/62.3/62.6/78.5 | **99.0/**80.1/**74.9/**89.2 | 98.7/58.8/60.0/91.8 | 97.5/58.9/59.3/86.9 |
| chewinggum | 98.8/54.5/**59.1**/82.6 | 98.7/48.0/48.5/81.2 | 98.4/19.5/35.7/**86.0** | 98.8/24.6/38.8/82.9 | **98.9/**43.0/44.2/81.1 | 98.4/**56.5**/56.2/80.6 |
| fryum | 96.6/44.2/49.5/90.3 | 97.1/43.4/51.5/73.8 | 91.1/37.9/46.0/85.0 | 95.0/**51.8/**52.3/70.2 | 92.6/38.3/45.9/89.1 | **97.0**/48.1/**52.9/91.1** |
| pipe_fryum | 99.0/51.0/55.5/93.7 | 99.3/56.9/63.7/90.7 | 98.9/59.6/60.4/91.5 | **99.7/89.2/82.6/**50.7 | 98.9/58.7/58.8/**96.4** | 98.9/51.1/54.7/94.0 |
| Mean | 98.6/39.3/43.8/91.7 | 98.4/31.6/37.4/85.3 | 98.1/36.8/41.2/88.7 | 98.5/**41.1/45.3/**71.6 | 98.5/40.8/43.7/91.1 | **98.8**/40.4/43.5/**91.9** |

