# OpenReview forum: "MambaAD: Exploring State Space Models for Multi-class Unsupervised Anomaly Detection"
_NeurIPS.cc/2024/Conference — NeurIPS 2024 poster_

### Official Review · Reviewer_wfnx · 2024-07-09

**Soundness:** 3
**Presentation:** 4
**Contribution:** 3
**Rating:** 8
**Confidence:** 5

**Summary:**

The paper pioneers the utilization of Mamba in the anomaly detection field. Specifically, the core contribution of the proposed MambaAD is a Locality-Enhanced State Space module, which utilizes Mamba for global information and CNN for local information. Whereas the motivation and the core technical contribution of this paper are similar to other works like U-Mamba, VMamba, and MedMamba, the proposed MambaAD is the first to extensively explore Mamba in the anomaly detection field and achieve state-of-the-art multi-class anomaly detection performance with a low parameter count and computational demand, making this paper acceptable for me.

**Strengths:**

This paper is the first to explore Mamba in the anomaly detection field.

The proposed MambaAD achieves state-of-the-art multi-class anomaly detection performance with lower computational demand and fewer parameters than other multi-class anomaly detection alternatives.

The experiments are extensive, featuring six distinct AD datasets with seven metrics.

**Weaknesses:**

1. The motivation for using Mamba for visual anomaly detection should be strengthened. e.g., what’s the core challenge for multi-class anomaly detection? why long-range modeling is important for multi-class anomaly detection?

2. It would be better to report the performance with only global/local branches in Table 5, which can better show the influence of individual branches.

3. While the proposed MambaAD achieves better detection performance in most metrics on the evaluated datasets, for some metrics, like AUPRO on VisA, the proposed MambaAD is not the best one. It would be better to offer some analysis on this issue.

4. The current MambaAD framework appears to still rely on a pre-trained CNN encoder, utilizing the Mamba model only in the decoder. Why is this design chosen? What would be the implications if the entire structure were based on the Mamba model?

5. It seems that only the Hilbert scanning method is ultimately used out of the five proposed hybrid scanning methods. How should we further interpret the ablation study in Table 3?

**Questions:**

See the weakness.

**Limitations:**

See the weakness.

---

> ### Author Rebuttal · Authors · 2024-08-03
>
> **Q1: Motivation**
>
> *(1)* Since the model needs to learn the **data distribution among samples with significant differences in a multi-class anomaly detection task, it requires a global modeling capability**, such as that provided by Transformers. UniAD, as the first model to propose a multi-class anomaly detection task, uses a Transformer decoder to model global features. However, **due to the quadratic computational complexity of Transformers, UniAD only uses Transformers on the smallest scale feature maps, which limits its performance in detecting small defects**.
>
> *(2)* Mamba, a model with **linear computational complexity**, is appropriately applied to multi-class anomaly detection. Its global modeling capability allows it to **better learn the data distribution of different category samples, and its linear computational complexity enables modeling at multiple scales, improving the detection performance for small defects**.
>
> In summary, Mamba's linear computational complexity and global modeling capability make it well-suited for addressing multi-class anomaly detection problems. As illustrated in Fig. 1 in the main text, MambaAD enhances its performance in multi-class anomaly detection by integrating CNNs with local modeling capabilities into the Mamba structure of the decoder, thereby improving its ability to learn the distribution of each feature.
>
> **Q2: Local/Global Branches**
>
> We conducted three ablation experiments to verify the impact of branches in the LSS module. The Local branch represents the use of only the parallel CNN branch, without the Mamba-based HSS branch. The Global branch represents the use of only the Mamba-based HSS branch, without the parallel CNN branch, making the decoder in this structure purely Mamba-based. Finally, Global+Local represents the proposed LSS structure used in MambaAD, which combines the serial Mamba-based HSS with parallel CNN branches of different kernel sizes.
> The experimental results are shown in the table below. The Local branch, which uses only CNNs, has the lowest parameter count and FLOPs but also the lowest mAD metric, indicating high efficiency but suboptimal accuracy. The Global method, based on the pure Mamba structure, consumes more parameters and FLOPs than the Local method but shows a significant improvement in performance **(+2.7%)**. Finally, the combined Global+Local method, which is the LSS module used in MambaAD, achieves the best performance with a notable improvement of **(+1.6%)** over the individual methods.
> | Method       | Params(M) | FLOPs(G) | mAD  |
> | ------------ | --------- | -------- | ---- |
> | Local        | 13.0      | 5.0      | 81.7 |
> | Global       | 22.5      | 7.5      | 82.1 |
> | Global+Local | 25.7      | 8.3      | **86.0**|
>
> **Q3: Evaluation Metrics**
>
> Please refer to **Q3** to **Reviewer gwhg** for more quantitative analysis in **Tab. 6** of the Rebuttal PDF and qualitative analysis in **Fig. 1** of the Rebuttal PDF.
>
> **Q4: Mamba Encoder**
>
> *(1)* Existing anomaly detection methods primarily use pre-trained encoders based on CNN backbone networks. For instance, [1,3] utilize WideResNet-50, while [2] employs EfficientNet-b4. Additionally, [4] uses the Transformer-based ViT as the feature extraction backbone. **Currently, no pre-trained encoder based on Mamba has been applied to the anomaly detection domain**. Therefore, MambaAD also uses widely adopted CNNs as the pre-trained backbone network.
>
> *(2)* In the main text, Tab. 2 presents ablation experiments on different CNN networks. The results show that ResNet-34 achieves a good balance between efficiency and performance. Although WideResNet-50 can achieve better results, its computational complexity and parameter count are eight to ten times higher than those of ResNet-34. Hence, MambaAD selects ResNet-34 as the backbone network.
>
> *(3)* To explore the impact of pre-trained Mamba encoders on anomaly detection tasks, we selected two recently popular vision Mamba models as backbone feature extraction networks: **VMamba [5] and EfficientVMamba [6]**. We replaced the original ResNet-34 backbone with these models, and the results are shown in the table below. The parameter count and FLOPs represent the values for the backbone network only, not the entire model, to facilitate the comparison of different backbone network performances. **EfficientVMamba-T has the fewest parameters and FLOPs but performs poorly on the MVTec-AD dataset in terms of the mAD metric. VMamba-T has a parameter count nearly four times that of ResNet-34, but its FLOPs are only 1.5 times higher. However, its mAD results are still inferior to those of ResNet-34.**
>
> Currently, existing pre-trained Mamba-based backbone networks are neither lightweight nor effective for feature extraction in anomaly detection. Therefore, **we will continue to research lightweight, efficient, and high-accuracy Mamba-based backbone networks and apply them as feature extractors in the anomaly detection domain.**
> | Backbone          | Params | FLOPs | mAD  |
> | ----------------- | ------ | ----- | ---- |
> | EfficientVMamba-T | 6.3M   | 1.0G  | 72.7 |
> | VMamba-T          | 30.2M  | 6.0G  | 82.9 |
> | ResNet34          | 8.2M   | 4.0G  | **86.0** |
>
> [1] Anomaly detection via reverse distillation from one-class embedding. Deng *et al.*. CVPR'22.
>
> [2] A unified model for multi-class anomaly detection. You *et al.*. NeurIPS'22.
>
> [3] Hierarchical vector quantized transformer for multi-class unsupervised anomaly detection. Lu *et al.*. NeurIPS'23.
>
> [4] Exploring plain vit reconstruction for multi-class unsupervised anomaly detection. Zhang *et al.*. arXiv'23.
>
> [5] Vmamba: Visual state space model. Liu *et al.*. arXiv'24.
>
> [6] Efficientvmamba: Atrous selective scan for light weight visual mamba. Pei *et al.*. arXiv'24.
>
> **Q5: Scanning Methods**
>
> Please refer to **Q1** to **Reviewer aMDx** and additional ablation experiments in **Tab. 5** of the Rebuttal PDF.

---

> > ### Comment · Reviewer_wfnx · 2024-08-08
> > **Response**
> >
> > After considering the author's responses to all reviewers during the rebuttal phase, I believe that the contribution and quality of this paper fully meet the high standards of NeurIPS. I strongly recommend accepting this paper and will modify my rate from 6 to 8. Here are my reasons:
> >
> > The authors are the first to apply Mamba to the field of anomaly detection, which represents a significant innovation. The introduction of the LSS and HSS modules demonstrates unique approaches and technical advantages in handling multi-class anomaly detection tasks. The experimental section is very detailed and convincing. The rebuttal provided extensive theoretical and experimental analyses that addressed most of my concerns.
> >
> > However, there is still some room for minor improvements. I suggest including the comparative experimental analysis with Transformer-based methods in the main text. Additionally, incorporating the pixel-level evaluation metrics in the main text would further substantiate the effectiveness of the approach in anomaly localization. Also, it will be interesting to study the influence of different input resolutions, since Mamba should have better long-term information acquisition capability.

---

> > > ### Author Response · Authors · 2024-08-13
> > > **Thanks and response to resolutions.**
> > >
> > > Dear Reviewer wfnx,
> > >
> > > **Thank you for recognizing our work and your valuable suggestions to help us improve the manuscript.** We commit to *including ablation experiments based on Transformer methods corresponding to Tab. 1, 7, and 8 in the revised version*. Additionally, we will incorporate the *mIoU evaluation metric results across six datasets to further validate the effectiveness of our approach in anomaly localization*.
> > >
> > > **Q: Different Input Resolutions**
> > >
> > > ***(1)*** Regarding the impact of different resolutions, particularly on long-term information acquisition capability, we compared MambaAD's performance on the **MVTec-AD and VisA datasets at $256^2$ and $512^2$ resolutions**, as shown in the table below. For the MVTec-AD and VisA datasets, the **RD4AD method shows a significant decline in all metrics as the resolution increases**, indicating that *purely convolutional approaches struggle with long-distance modeling*.
> > >
> > > ***(2)*** For the Transformer-based UniAD method, increasing the resolution on the MVTec-AD dataset results in a **decline in image-level metrics, while pixel-level metrics such as P-AP, P-F1_max, P-AUPRO, and P-mIoU improve**. A similar trend is observed in MambaAD. This outcome may be due to the enhanced and clearer detailed information in higher resolutions, which **aids the model in more accurately identifying and distinguishing different regions for pixel-level tasks**, thereby improving anomaly localization metrics. However, for image-level classification tasks, **excessive detail may introduce noise and increase model complexity, leading to a decline in performance**. The decrease in the P-AUROC metric could be attributed to AUROC's nature as a measure of a classifier's overall performance across different thresholds, *reflecting the model's ability to distinguish between positive and negative samples*. *Higher resolution might cause instability in the model's performance at certain thresholds, especially for borderline samples, leading to a decrease in AUROC*. Overall, on the MVTec-AD dataset, the **MambaAD model outperforms both the CNN-based RD4AD and the Transformer-based UniAD at $512^2$ resolution**.
> > >
> > > ***(3)*** On the VisA dataset, the **UniAD method shows a general decline in performance at $512^2$ resolution compared to $256^2$ resolution**. This could be due to the *smaller anomaly areas in the VisA dataset, where Transformers may introduce noise during high-resolution global modeling and lack local region modeling capability*. For the **MambaAD** method, **increasing the resolution significantly improves all metrics except P-AUROC**. This improvement is attributed to the **effective long-distance modeling capability of Mamba and the integration of the proposed Locality-Enhanced LSS module. *The decline in P-AUROC might be due to changes in the distribution of positive and negative samples caused by the increased detail, affecting AUROC*. Overall, on the **VisA** dataset, **increasing the resolution significantly enhances MambaAD's performance compared to CNN and Transformer-based models**.
> > >
> > > In the future, we will continue to explore the impact of different resolutions, particularly higher resolutions such as $1024^2$ and $2048^2$, to demonstrate the effectiveness of the Mamba model in high-resolution anomaly detection.
> > >
> > > ***MVTec-AD Dataset***
> > > | Resolution | Method  | I-AUROC | I-AP   | I-F1_max | P-AUROC | P-AP   | P-F1_max | P-AUPRO | P-mIoU |
> > > | ---------- | ------- | ------ | ---- | ------ | ------ | ---- | ------ | ------ | ---- |
> > > | $256^2$ | RD4AD | 94.6   | 96.5 | 95.2   | 96.1   | 48.6 | 53.8   | 91.1   | 37.0 |
> > > | $256^2$ | UniAD   | 96.5   | 98.8 | 96.2   | 96.8   | 43.4 | 49.5   | 90.7   | 32.5 |
> > > | $256^2$ | MambaAD | 98.6   | 99.6 | 97.8   | 97.7   | 56.3 | 59.2   | 93.1   | 41.2 |
> > > |$512^2$| RD4AD | 86.0   | 91.9 | 90.3   | 92.9   | 45.4 | 49.1   | 88.2   | 33.5 |
> > > |$512^2$ | UniAD   | 96.3   | 98.5 | 95.5   | 95.6   | 46.9 | 50.1   | 91.5   | 33.7 |
> > > | $512^2$ | MambaAD | 97.7   | 99.0 | 96.3   | 96.8   | 60.4 | 60.9   | 93.1   | 44.5 |
> > >
> > > ***VisA Dataset***
> > > | Resolution | Method  | I-AUROC | I-AP   | I-F1_max | P-AUROC | P-AP   | P-F1_max | P-AUPRO | P-mIoU |
> > > | ---------- | ------- | ------ | ---- | ------ | ------ | ---- | ------ | ------ | ---- |
> > > | $256^2$ | RD4AD | 92.4   | 92.4 | 89.6   | 98.1   | 38.0 | 42.6   | 91.8   | 27.9 |
> > > |$256^2$ | UniAD   | 88.8   | 90.8 | 85.8   | 98.3   | 33.7 | 39.0   | 85.5   | 25.7 |
> > > | $256^2$| MambaAD | 94.3   | 94.5 | 89.4   | 98.5   | 39.4 | 44.0   | 91.0   | 29.5 |
> > > | $512^2$| RD4AD | 89.1   | 90.2 | 86.9   | 93.8   | 29.8 | 37.0   | 89.7   | 23.2 |
> > > | $512^2$| UniAD   | 87.4   | 88.5 | 84.8   | 96.6   | 26.7 | 35.2   | 87.2   | 22.0 |
> > > | $512^2$| MambaAD | 95.7   | 97.0 | 92.5   | 96.1   | 40.1 | 46.1   | 92.0   | 31.0 |
> > >
> > > Best regards!
> > >
> > > Authors of MambaAD.

---

### Official Review · Reviewer_gwhg · 2024-07-13

**Soundness:** 3
**Presentation:** 3
**Contribution:** 3
**Rating:** 5
**Confidence:** 4

**Summary:**

This paper employs a pyramid-structured auto-encoder to reconstruct multi-scale features, utilizing a pre-trained encoder and a decoder based on the Mamba architecture. The experimental results show SoTA performances on several commonly used datasets.

**Strengths:**

1. As far as I am concerned, MambaAD is the first attempt to use Mamba in the anomaly detection domain.
2. The experimental results are SOTA.
3. The representation is clear and easy to follow.

**Weaknesses:**

1. My major concern is the efficiency comparison shown in Table 4. UniAD is a transformer-based method, but leading to the smallest params and FLops. The advantage of Mamba should be the efficiency, however, which is not as good as transformer-baed methods.
2. The backbone model is WideResNet50. The backbone model of the other method is not WideResNet50. For example, UNIAD use the efficient-net. I am not sure about wether the improvements in this model imply in different backbone model.

**Questions:**

1. Why the params and FLOPs are higher than the transformer-based model?
2. Why the evaluation metrics on the pixel level is lower? Is the anomaly localization worse?

**Limitations:**

The limitation and broader societal impacts are mentioned in the last paragraph.

---

> ### Author Rebuttal · Authors · 2024-08-03
>
> **Q1: Compared with Transformer-based Method**
>
> *(1)* As shown in Fig. 1 of the main text, although UniAD is a Transformer-based method, it employs a **single-scale approach**, whereas our MambaAD, based on Mamba, utilizes a **multi-scale approach**. Specifically, **single-scale methods model and reconstruct only the smallest scale feature map during the decoder stage**. In contrast, multi-scale methods progressively increase the resolution from the smallest scale feature map during the decoder stage, modeling and reconstructing feature maps at multiple scales. **Due to the quadratic computational complexity of Transformers, UniAD opts for a single-scale framework as its primary architecture.** Consequently, UniAD, which reconstructs only the smallest scale feature map, naturally has the least parameter count and FLOPs. On the other hand, MambaAD, which models feature maps at three different scale resolutions, has slightly higher parameter counts and FLOPs compared to UniAD.
>
> *(2)* To ensure a fair comparison, **we used the same encoder, ResNet-34**, as the feature extraction network. **Within the current multi-scale framework** of MambaAD, we employed both a pure Transformer-based decoder and a pure Mamba-based decoder to validate the efficiency of the method. The results are shown in the table below. It can be observed that, **under the same framework**, the pure Mamba-based decoder has significantly fewer parameters **-4.6M** and FLOPs **-0.9G** compared to the pure Transformer-based decoder, demonstrating a clear efficiency advantage. Additionally, in terms of performance, the Mamba-based method shows a notable improvement in the mAD metric (**+1.9%**), achieving a significant advantage in effectiveness as well.
> |Decoder| Params(M)|FLOPs(G)|mAD|
> |---|---|---|---|
> |Transformer-based|27.1|8.4|80.2|
> |Mamba-based|22.5|7.5|**82.1**|
>
> **Q2: Different Backbone**
>
> *(1)* The backbone feature extraction network used in MambaAD is **ResNet-34**. Although WideResNet-50 can achieve slightly better performance (+0.6%), it has **ten times the number of parameters and more than eight times the computational complexity**. Therefore, we opted for the more **lightweight ResNet-34 as the backbone**.
>
> *(2)* To ensure a fairer comparison with UniAD and eliminate the influence of different backbone networks, we replaced the backbone networks in UniAD with three different options: **EfficientNet-b4, ResNet-34, and WideResNet-50**. Similarly, we replaced the backbone networks in MambaAD with these three options. Originally, UniAD with an EfficientNet-b4 has 3.6G FLOPs and an accuracy of 81.7. When we replaced the MambaAD backbone with EfficientNet-b4, it achieved only **2.7G** FLOPs and an accuracy of 84.1, which is **+2.4%** higher than UniAD. **With the same backbone network, MambaAD has fewer parameters and FLOPs and better performance compared to UniAD**. Subsequently, replacing the UniAD backbone with ResNet-34 resulted in a performance decline, while further replacing it with WideResNet-50 led to a +0.7% improvement in performance, albeit with doubled parameters and FLOPs.
>
> In conclusion, **under the same backbone network, MambaAD consistently outperforms the Transformer-based single-scale UniAD method**, demonstrating significant improvements in performance.
> |UniAD | Params(M) | FLOPs(G) | mAD |
> |---|---|---|---|
> |EfficientNet-b4|24.5|3.6|81.7|
> |ResNet-34|16.1|6.1|78.2|
> |WideResNet-50|33.5|14.4|78.9|
>
> |MambaAD|Params(M)|FLOPs(G)|mAD|
> |---|---|---|---|
> |EfficientNet-b4|23.6|2.7| 84.1 |
> |ResNet-34|25.7|8.3|86.0|
> |WideResNet-50|268|68.1| 86.6 |
>
> **Q3: Evaluation Metrics**
>
> *(1)* From the perspective of method categorization, our method and the compared SoTA methods can be distinguished as follows: RD4AD, UniAD, DiAD, and MambaAD are **rec.-based** anomaly detection methods. DeSTSeg, on the other hand, is a **hybrid method that combines data augmentation and reconstruction**. During training, **DeSTSeg artificially introduces random anomalous regions to construct anomalous data, enabling the segmentation network to learn the precise locations of these regions**, thereby reducing the probability of missed and false detections. In contrast, other **rec.-based methods only learn the feature distribution of normal samples and perform comparisons at the feature level during testing**. Consequently, the hybrid DeSTSeg method may outperform other rec.-based methods in certain anomaly localization metrics. However, MambaAD, with its simpler approach, **achieves the majority of SoTA metrics and a few near-optimal results overall**.
>
> *(2)* When comparing rec.-based methods, MambaAD demonstrates its effectiveness and robustness across four anomaly localization evaluation metrics on **six different datasets**. Recent work [1] has suggested that existing anomaly localization evaluation standards are **not specifically designed for segmentation-based anomaly detection tasks**. We selected the most commonly used **IoU metric to demonstrate the accuracy of anomaly localization which intuitively reflects the accuracy and overlap**. We tested the **mIoU** of rec.-based SoTA methods on six different types of six datasets. The results, as shown in **Tab. 6** in Rebuttal PDF, indicate that MambaAD significantly outperforms existing SoTA methods in terms of mIoU values across all six datasets, thereby proving the accuracy and robustness of our method in anomaly localization.
>
> Finally, we conduct a **qualitative analysis** of the MambaAD method in comparison to RD4AD within the same framework. As visualized in **Fig. 1** in Rebuttal PDF, MambaAD demonstrates higher localization accuracy on anomaly images compared to RD4AD. This further validates the accuracy of the method in anomaly localization.
>
> [1] Learning Feature Inversion for Multi-class Anomaly Detection under General-purpose COCO-AD Benchmark. Zhang *et al.*. arXiv'24.

---

> > ### Comment · Reviewer_gwhg · 2024-08-11
> >
> > Thanks for your detailed response. Some of my issues have been addressed. But why does the MambaAD with the backbone ResNet-34 and WideResNet-50 result in more parameters and Flops (compared to UNIAD)? Moreover, could you please list the inference time or FPS to show the inference speed of MambaAD and UniAD?

---

> > > ### Author Response · Authors · 2024-08-11
> > > **Response to Reviewer gwhg about FLOPs and FPS**
> > >
> > > Dear Reviewer gwhg:
> > >
> > > Thank you for your comments.
> > >
> > > **Q: FLOPs**
> > >
> > > (1) As shown in Fig. 1 of the main text, UniAD is a single-scale method, whereas our Mamba-based MambaAD is a multi-scale method. UniAD **concatenates features from the four stages of the encoder** and uses a Transformer to model **only the smallest scale feature map**. In contrast, MambaAD gradually **increases the resolution of the feature maps** during the decoder stage and performs modeling and reconstruction at four different scales. This difference in framework leads to MambaAD consuming more parameters and FLOPs **on high-resolution feature maps**. Additionally, the WideResNet-50 backbone network, with its higher number of channels at each scale, further increases the parameter and FLOP differences between the two methods.
> > >
> > > (2) The Mamba-based model offers **better interpretability and performance**. The model's FLOPs are also related to the implementation method. For instance, Mamba's **FLOPs could be reduced if the state transition is implemented using a vanilla for-loop**. However, the authors of Mamba chose to double the FLOPs to achieve a more parallel and efficient implementation.
> > >
> > > **Q: FPS**
> > >
> > > We evaluated the FPS of UniAD and MambaAD on ResNet-34 and WideResNet-50 backbones to demonstrate inference speed. The results show that MambaAD still lags behind UniAD in terms of inference speed. The following are possible reasons and directions for optimization:
> > >
> > > (1) Model framework: As mentioned in ***Q: FLOPs***, the MambaAD model requires more FLOPs, which affects inference speed to some extent.
> > >
> > > (2) Code implementation: The MambaAD code currently uses the initial Mamba implementation with mamba_ssm and causal-conv1d for SSM, which suffers from suboptimal algorithmic efficiency. Using the latest CUDA-accelerated selective_scan_cuda_core algorithm could improve computational efficiency by **40%**.
> > >
> > > (3) Number of scanning directions: The current method uses eight different scanning directions to ensure optimal performance. Using only **two different scanning directions** on the ResNet-34 backbone can still achieve an **mAD of 85.8 and an FPS of 574.6**, representing a **258% increase in FPS** compared to using eight scanning directions.
> > >
> > > (4) Code implementation of scanning methods: The current Hybrid Scan method is implemented solely in PyTorch, resulting in low efficiency. Our observations indicate that the HS encoder and HS decoder **occupy more than 60% of the time**. We plan to use **Triton or CUDA to accelerate** this process in the future.
> > >
> > > In summary, while MambaAD shows **significant performance improvements** over UniAD, its inference speed is affected by the aforementioned issues. We will continue to optimize the model and algorithms to surpass the inference speed of Transformer-based frameworks.
> > >
> > > | UniAD | Params(M) | FLOPs(G) | FPS | mAD |
> > > | ------------ | --------- | ----- | ---- | ---- |
> > > | ResNet-34 | 16.1 | 6.1 | 1119.9 | 78.2 |
> > > | WideResNet-50 | 33.5 | 14.4 | 582.7 | 78.9 |
> > >
> > > | MambaAD | Params(M) | FLOPs(G) | FPS | mAD  |
> > > | ----------- | --------- | ------ | ---- | ---- |
> > > | ResNet-34 | 25.7 | 8.3 |  222.5 | 86.0 |
> > > | WideResNet-50 | 268  | 68.1 | 44.6 | 86.6 |
> > >
> > >
> > > Best regards!
> > >
> > > Authors of MambaAD.

---

### Official Review · Reviewer_aMDx · 2024-07-13

**Soundness:** 3
**Presentation:** 3
**Contribution:** 3
**Rating:** 7
**Confidence:** 3

**Summary:**

The paper "MambaAD: Exploring State Space Models for Multi-class Unsupervised Anomaly Detection" introduces MambaAD, a novel framework for multi-class unsupervised anomaly detection using Mamba-based models. The framework consists of a pre-trained encoder and a Mamba decoder that integrates Locality-Enhanced State Space (LSS) modules at multiple scales. The LSS module combines Hybrid State Space (HSS) blocks and multi-kernel convolution operations to capture both long-range and local information. The proposed method is evaluated on six diverse anomaly detection datasets, demonstrating state-of-the-art performance and significant improvements in efficiency and accuracy.

**Strengths:**

1. The use of Mamba-based models for multi-class anomaly detection and the innovative LSS module are novel contributions.
2. The theoretical foundations are robust, and the empirical validation is comprehensive.
3. The paper is clearly written, with well-structured sections and effective use of visual aids.
4. The approach addresses a critical gap in anomaly detection research and demonstrates significant performance improvements, making it highly relevant and impactful.

**Weaknesses:**

1. The sensitivity of the method to the selection of scanning methods and directions could be explored further.
2. While the experimental results are compelling, additional validation on more diverse and challenging datasets would further strengthen the claims.
3. The discussion of potential limitations and future work could be expanded to provide a more comprehensive view of the method's applicability and areas for improvement.

**Questions:**

1. Could the authors provide more insights into the selection of scanning methods and the impact of different numbers of directions on the detection performance?
2. Have the authors considered the robustness of the method under different environmental conditions, such as varying lighting and occlusions?
3. Could additional experiments on other industrial or medical datasets help to further validate the generalizability of the proposed method?

===== post rebuttal ======

The authors' rebuttal solve most of my concerns, hence I raise my score to 7.

**Limitations:**

The authors have adequately addressed some limitations, including potential issues with scanning method selection and environmental conditions. Constructive suggestions for improvement include exploring the sensitivity to scanning parameters and validating the method under different environmental conditions.

---

> ### Author Rebuttal · Authors · 2024-08-03
>
> **Q1: Further Exploration of Scanning Methods and Direction Selection**
>
> ***(1) Scan Directions***
>
> Firstly, we further investigate the impact of different scanning directions on Params, FLOPs, training time over 100 epochs, training memory usage, and final performance, as shown in the table below. It is evident that as the number of scanning directions increases, there is a corresponding increase in Params, FLOPs, training time, and training memory usage. Consequently, the performance in terms of mAD also improves. Therefore, for those prioritizing efficiency, scanning with **2 directions offers the lowest Params and FLOPs, while also requiring less training time and memory, yet achieving satisfactory performance**. Conversely, for those aiming for optimal performance and willing to accept increased computational load and training time, scanning with **8 directions yields the best results**.
> | Scan Directions | Params(M) | FLOPs(G) | Training Time | Training Memory | mAD  |
> | --------------- | --------- | -------- | ------------- | --------------- | ---- |
> | 2               | 21.8     | 7.2     | 2h1m          | 6230           | 85.8 |
> | 4               | 23.1    | 7.5     | 2h56m         | 7974           | 85.9 |
> | 8               | 25.7     | 8.3     | 5h39m         | 11802          | 86.0 |
>
> ***(2) Scan Methods***
>
> Subsequently, we further explore the impact of different scanning methods and the fusion of multiple scanning methods on the results, as shown in the table below. Different scanning methods exhibit negligible differences in terms of parameters, FLOPs, training time, and memory usage, with variations only observed in the final results. The results indicate that using any of the five different scanning methods individually can achieve satisfactory outcomes. However, combining different scanning methods tends to degrade performance, possibly due to the significant differences between the methods, which may hinder the selective scanning and global modeling convergence of the SSM. Therefore, we ultimately choose to use only the Hilbert scanning method. **This method allows for continuous scanning in the central part of the image, which is advantageous for SSM in learning and modeling the distribution of anomaly detection data, as most data in anomaly detection datasets are centered in the image.**
> |Sweep|Scan|Zorder|Zigzag|Hilbert|mAD|
> |-----|----|------|-----|-----|----|
> |√|||||85.8|
> ||√||||85.9|
> |||√|||85.8|
> ||||√||85.8|
> |||||√|86.0|
> ||||√|√|84.9|
> |||√||√|85.3|
> ||√|||√|85.4|
> |√||||√|85.7|
> ||√|√|√|√| 85.3 |
>
> **Q2: Additional Datasets**
>
> In Tab. 1 of the main text, we demonstrate the effectiveness and robustness of our method on three industrial anomaly detection datasets: MVTec-AD, VisA, and Real-IAD. To further validate the effectiveness of our method, we include additional datasets from **medical, 3D, and real-world scenes**. The **Uni-Medical** dataset [1] comprises CT scans from three different anatomical locations. The **MVTec-3D** dataset [2] includes both RGB images and depth maps. Additionally, [3] extends the commonly used COCO dataset for detection and segmentation to anomaly detection, creating a large-scale, general-purpose **COCO-AD** dataset that encompasses complex scenes and diverse distributions. These datasets exhibit significant distributional differences compared to traditional industrial anomaly detection datasets, providing a robust test for our method. The results are summarized in the table below. For more detailed results, please refer to **Appendix C-E**. Our MambaAD method shows improvements over existing SoTA methods, with **+4.2%** on the medical dataset, **+3.7%** on the 3D dataset, and **+2.5%** on the COCO-AD dataset. These significant improvements across datasets with different distributions and scenarios validate the effectiveness and robustness of our method.
> | Extra Datasets | RD4AD | UniAD | SimpleNet | DeSTSeg | DiAD | MambaAD |
> | --------------- | ----- | ----- | --------- | ------- | ---- | ------- |
> |Uni-Medical| 70.2  | 69.9  | 69.1| 60.9| 70.5 | **74.7**|
> |MVTec-3D| 74.3  | 71.1  | 66.9| 69.1| 73.8 | **78.0**|
> |COCO-AD| 45.0  | 42.3  | 40.7| 40.0| 44.1 | **47.5**|
>
> **Q3: Robustness Under Different Environmental Conditions**
>
> **Currently, the anomaly detection field lacks specialized datasets that account for varying lighting conditions and occlusions.** However, the **COCO-AD[3] dataset** we use is a general-purpose anomaly detection dataset derived **from real-world detection and segmentation scenarios**. It includes a wide variety of objects and categories from real-world settings. As illustrated in Fig. 1(a) of the [3] paper, the dataset encompasses diverse scenes with significant variations between them. These scenes exhibit different data distributions and **include various environmental conditions such as sunny and cloudy days**. Additionally, the dataset features **occlusions both within the same object and between different objects**. In the table above, our method achieves SoTA on the COCO-AD, with an improvement of over **+2.5%** in the mAD metric. This further validates the robustness of our method in complex and general anomaly detection scenarios.
>
> [1] Exploring plain vit reconstruction for multi-class unsupervised anomaly detection. Zhang *et al.*. arXiv'23.
>
> [2] The mvtec 3d-ad dataset for unsupervised 3d anomaly detection and localization. Bergmann *et al.*. arXiv'21.
>
> [3] Learning Feature Inversion for Multi-class Anomaly Detection under General-purpose COCO-AD Benchmark. Zhang et al.. arXiv'24.
>
> **Q4: Limitations and Future Work**
>
> The primary limitations lie in **high-resolution application scenarios** and the **Mamba-based encoder**. These areas will be the focus of future research. For a more detailed explanation, please refer to **Q1** addressed to Reviewer **FXx2**.

---

> ### Author Response · Authors · 2024-08-11
> **Thanks.**
>
> Dear reviewer aMDx:
>
> Thank you for recognizing our work. We commit to incorporating the content you suggested in the revised version. Thank you again for your effort in the review and the discussion!
>
> Best regards!
>
> Authors of MambaAD.

---

### Official Review · Reviewer_FXx2 · 2024-07-17

**Soundness:** 2
**Presentation:** 2
**Contribution:** 1
**Rating:** 5
**Confidence:** 5

**Summary:**

This paper introduces a method for multi-class unsupervised anomaly detection utilizing a CNN-based encoder and a Mamba-based decoder. It incorporates an LSS module and a HSS block within the Mamba decoder to enhance performance. The effectiveness of the method is validated through experiments conducted on three image anomaly detection datasets.

**Strengths:**

1. This study represents a novel exploration in using Mamba for anomaly detection. As far as I understand, this is the first research endeavor to apply Mamba in the context of anomaly detection.
2. Extensive experiments are presented in the paper.

**Weaknesses:**

1. From a methodological standpoint, the anomaly detection approach essentially implements RD4AD, with the CNN decoder replaced by a Mamba decoder.
2. Several key ablations are absent from the experimentation:
(a) It would be nice to conduct a comparison study using the same RD4AD architecture with CNN-based decoder, transformer-based decoder, and Mamba-based decoder. This comparison would validate the effectiveness of Mamba compared to other decoder architectures.
(b) Ablations of model components should be included, which compares the proposed method, the proposed method without the HSS block, and the proposed method without the LSS module. This ablation study would provide insights into the functionality of these components.

**Questions:**

Please refer to the comments in the Weakness Section.
---------------
Post-rebuttal review:
While I still have some reservations about the methodological novelty of this study, the performance and generalizability of the method appear to be strong. After reviewing the authors' response and considering the feedback from other reviewers, I have decided to raise my initial rating.

**Limitations:**

Minimal discussion. Although the section labeled "limitations" includes some wording, it does not actually discuss any perspectives on real limitations on the methods.

---

> ### Author Rebuttal · Authors · 2024-08-06
>
> **Q1: From a methodological perspective**
>
> *(1)* We adopt a **reconstruction-based framework**. Anomaly detection methods based on reconstruction can be broadly categorized into **image reconstruction and feature reconstruction**. Methods such as GAN and diffusion-based approaches primarily fall under image reconstruction, where the similarity between the generated image and the input is calculated. In contrast, feature reconstruction methods compute the similarity between the input features and the reconstructed features. **RD4AD is merely a specific instance within the feature reconstruction methods. Additionally, this framework is not referred to as a contribution in our paper.**
>
> *(2)* Mamba, as a novel global modeling structure, **has not yet been explored for its potential in anomaly detection (AD)**. We found that directly applying it to multi-class anomaly detection tasks does not significantly improve performance compared to the original CNN architecture (+0.4%) in Tab. 1. Therefore, we first combined Mamba's global modeling capability with CNN's local modeling capability in Tab. 3, which significantly improved performance (**+2.8%**), and proposed the **AD-adaptive LSS module**, corresponding to the contribution in the fusion modeling approach. Secondly, we introduced the **AD-adaptive HSS method** to specifically enhance the scanning approach (**+1.1%**) for multi-class anomaly detection data, corresponding to the contribution in the scanning method.
>
> **Q2: More key ablations**
>
> ***(1) Decoder Ablations***
>
> We conducted experiments on different decoder structures **within the same RD4AD framework**, and the results are summarized in the table below. The Mamba-based decoder, which uses only Mamba as the component for each decoder module instead of the proposed LSS module that integrates Mamba with CNN structures, shows a decrease in performance compared to the result of 86.0 reported in the paper. However, compared to decoders based purely on CNN or Transformer architectures, the Mamba-based decoder achieves the highest mAD score of **82.1**. Additionally, **the Mamba-based decoder has fewer parameters and FLOPs compared to the Transformer-based structure.**
> | Decoder | Params(M) | FLOPs(G) | mAD |
> | --- | --- | --- | --- |
> | CNN-based | 13.0 | 5.0 | 81.7 |
> | Transformer-based | 27.1 | 8.4 | 80.2 |
> | Mamba-based | 22.5 | 7.5 | 82.1 |
>
> ***(2) Components Ablations***
>
> The ablation experiments for the proposed components are summarized in the table below. Using the most **basic decoder purely based on Mamba and employing only the simplest two-directional sweep scanning method**, we achieve an mAD score of 82.1 on the MVTec-AD dataset. Subsequently, by incorporating the proposed **LSS module**, which integrates the global modeling capabilities of Mamba with the local modeling capabilities of CNNs, the mAD score improves by **+2.8%**. Finally, replacing the original scanning directions and methods with **HSS**, which combines features from different scanning directions and employs the **Hilbert scanning method**, better aligns with the data distribution in most industrial scenarios where objects are centrally located in the image. This results in an additional **+1.1%** point improvement in the mAD score. Overall, the proposed MambaAD achieves an mAD score of **86.0** on the MVTec-AD dataset, reaching the SoTA performance.
> | Basic Mamba Decoder | LSS | HSS | mAD |
> | --- | --- | --- | --- |
> | ✓ |  |  | 82.1 |
> | ✓ | ✓ |  | 84.9 |
> | ✓ | ✓ | ✓ | 86.0 |
>
> **Q3: More Limitations**
>
> *(1)* Although Mamba has linear computational complexity, the current multi-scale reconstruction-based framework (*c.f.* Fig. 1) is more effective than the single-scale UniAD method (81.7 -> 86.0). However, due to the limitations of the framework, existing methods are still not efficient enough.
>
> *(2)* Moreover, since most industrial anomaly defects are relatively small, as seen in datasets like VisA and Real-IAD, where defects occupy a very low proportion of the entire image, it is necessary to improve detection resolution to identify small target defects. However, increasing the resolution brings additional computational complexity to the model. Therefore, **designing more efficient and lightweight anomaly detection models is urgently needed in industrial scenarios.** We extended the MambaAD method to high-resolution tasks, such as $512^2$, $1024^2$, and even higher $2048^2$ input resolutions. At a resolution of $512^2$, the model's FLOPs reach 33.2G with a throughput of only 45.1. At a resolution of $1024^2$, the FLOPs soar to 133G with a throughput of only 10.3. At a resolution of $2048^2$, we could not measure the corresponding FLOPs due to OOM (Out of Memory) errors. In summary, **there is still much room for improvement in the lightweight design of current models, especially when applied to high-resolution industrial scenarios. The current models do not yet meet the efficiency and real-time requirements of high-resolution industrial applications.**
>
> *(3)* **The existing Mamba-based encoder still has certain deficiencies in terms of efficiency and effectiveness**. In the table below, we replaced the current encoder with two different lightweight pre-trained Mamba encoders. The parameters and FLOPs only represent the results of the encoder. Although EfficientVMamba-T has the highest efficiency, its feature extraction capability is lacking. On the other hand, VMamba-T shows significant improvement in effectiveness but still lags behind the current CNN encoder in terms of real-time performance and overall effectiveness. Therefore, **designing more lightweight, efficient, and accurate Mamba-based encoders for anomaly detection is crucial and will be the focus of our future research.**
>
> | Backbone| Params(M) | FLOPs(G) | mAD  |
> | ------- | ------ | ----- | ---- |
> | EfficientVMamba-T | 6.3   | 1.0  | 72.7 |
> | VMamba-T| 30.2  | 6.0  | 82.9 |
> | ResNet34| 8.2   | 4.0  | 86.0 |

---

> > ### Comment · Reviewer_FXx2 · 2024-08-11
> > **Thanks for the rebuttal**
> >
> > I appreciate the authors' rebuttal and their engagement with my comments. Given that the experiments were conducted on multiple datasets, could the authors clarify which specific dataset was used to obtain the ablation mAD in the table above?

---

> > > ### Author Response · Authors · 2024-08-11
> > > **Response to Reviewer FXx2 about mAD.**
> > >
> > > Dear reviewer FXx2:
> > >
> > > Thank you for your comments. The ablation experiments in the three tables mentioned in the rebuttal were all conducted on the **MVTec-AD** dataset. **In the PDF, only Tab. 4 and 6** include experiments conducted on **multiple datasets**, as indicated in the first row of the respective tables. **All other tables' ablation experiments** were performed on the standard **MVTec-AD dataset**.
> > >
> > > Best regards!
> > >
> > > Authors of MambaAD.

---

> > > > ### Author Response · Authors · 2024-08-12
> > > > **Supplementary Ablation Experiments on VisA dataset.**
> > > >
> > > > **Q: Dataset Ablations**
> > > >
> > > > To further validate the reliability of the ablation experiments, we have added ablation experiments on the **VisA dataset on mAD** evaluation metric. Due to time constraints, we only conducted ablation experiments on **Tab. 1, 2, 3, 9, and 10 from the PDF on the VisA dataset**. The results on the **VisA dataset are consistent with the conclusions drawn from the MVTec-AD dataset**. We will provide more detailed and comprehensive experiments as supplementary material in the final version.
> > > >
> > > >
> > > > ***Tab. 1 Decoder Ablations***
> > > > | Method                    | Params | FLOPs | MVTec-AD | VisA |
> > > > | ------------------------- | --------- | -------- | -------- | ---- |
> > > > | CNN-based       | 13.0M        | 5.0G        | 81.7     | 72.5 |
> > > > | Transformer-based  | 27.1M      | 8.4G      | 80.2     | 67.4 |
> > > > | Mamba-based        | 22.5M      | 7.5G      | 82.1     | 72.9 |
> > > >
> > > >
> > > > ***Tab. 2 Scaning Directions Ablations***
> > > > | Scan Directions | Params | FLOPs | Training Time | Training Memory | MVTec-AD | VisA |
> > > > | --------------- | --------- | -------- | ------------- | --------------- | -------- | ---- |
> > > > | 2               | 21.8M     | 7.2G     | 2h1m          | 6230            | 85.8     | 78.5 |
> > > > | 4               | 23.1M     | 7.5G     | 2h56m         | 7974            | 85.9     | 78.6 |
> > > > | 8               | 25.7M     | 8.3G     | 5h39m         | 11802           | 86.0       | 78.9 |
> > > >
> > > > ***Tab. 3 Components Ablations***
> > > > | Basic Mamba | LSS | HSS | MVTec-AD | VisA |
> > > > | ------------------- | --- | --- | -------- | ---- |
> > > > | √                   |     |     | 82.1     | 72.9 |
> > > > | √                   | √   |     | 84.9     | 78.0   |
> > > > | √                   | √   | √   | 86.0       | 78.9 |
> > > >
> > > > ***Tab. 9 Branches Ablations***
> > > > | Method       | Params | FLOPs | MVTec-AD | VisA |
> > > > | ------------ | --------- | -------- | -------- | ---- |
> > > > | Local        | 13.0M      | 5.0G      | 81.7     | 72.5 |
> > > > | Global       | 22.5M      | 7.5G      | 82.1     | 72.9 |
> > > > | Global+Local | 25.7M      | 8.3G      | 86.0     | 78.9 |
> > > >
> > > > ***Tab. 10 Mamba-based Backbone Ablations***
> > > > | Backbone          | Params | FLOPs | MVTec-AD | VisA |
> > > > | ----------------- | ------ | ----- | -------- | ---- |
> > > > | EfficientVMamba-T | 6.3M   | 1.0G  | 72.7     | 71.2 |
> > > > | VMamba-T          | 30.2M  | 6.0G  | 82.9     | 76.2 |
> > > > | ResNet34          | 8.2M   | 4.0G  | 86.0     | 78.9 |

---

> > > > > ### Author Response · Authors · 2024-08-13
> > > > > **Please let us know whether we address all the issues**
> > > > >
> > > > > Dear reviewer,
> > > > >
> > > > > Thank you for the comments on our paper.
> > > > >
> > > > > We have submitted the response to your comments. Please let us know if you have additional questions so that we can address them during the discussion period. We hope that you can consider raising the score after we address all the issues.
> > > > >
> > > > > Thank you!
> > > > >
> > > > > Best Regards!
> > > > >
> > > > > Authors of # Mamba AD

---

### Author Rebuttal · Authors · 2024-08-06

**We would like to express our sincere gratitude for the constructive comments and suggestions from the reviewers. We appreciate the efforts of all reviewers in evaluating and helping to improve the quality of our manuscript.**

***The primary contributions of this paper are as follows:***

This paper is the **first to explore the application of Mamba in multi-class unsupervised anomaly detection**, proposing MambaAD. Existing CNN-based methods are limited by their long-distance modeling capabilities, while Transformer-based methods are constrained by their quadratic computational complexity. Therefore, this paper introduces the Locality-Enhanced State Space (**LSS**) module, which includes parallel cascaded Hybrid State Space (**HSS**) blocks and multi-kernel convolution operations. This module **integrates Mamba's linear computational complexity for global modeling with CNN's locally enhanced modeling capabilities at multiple scales in the decoder**. The HSS block employs Hybrid Scanning (**HS**) to map features into **five different scanning patterns and eight different scanning directions**, enhancing the global modeling capability of the State Space Model (SSM).

The effectiveness and robustness of MambaAD are validated across **six different datasets**: MVTec-AD, VisA, Real-IAD (**industrial**), Uni-Medical (**medical**), MVTec-3D (**3D**), and COCO-AD (**general scenarios**). The evaluation is conducted using **seven metrics at the image level (AUROC, AP, F1-Max) and pixel level (AUROC, AP, F1-Max, AUPRO)**. Additionally, during the Rebuttal phase, the **mIoU metric** for anomaly localization was included, making a total of **eight evaluation metrics**. The results demonstrate that MambaAD achieves **SoTA performance**. The **efficiency** of the method is also validated in both the main text and the Rebuttal. This work provides a foundation for further research on the application of Mamba in anomaly detection.

***Below is a brief summary of our responses to all reviewers' questions, along with their corresponding relationships to the tables and figure in the PDF.***

### Reviewer FXx2
- **Q1:** We discuss the **methodology** of MambaAD and its relationship with the RD4AD method.
- **Q2:**
  - We conducted additional ablation experiments on the **decoder**, corresponding to Tab. 1 in the PDF, demonstrating the efficiency and effectiveness of the Mamba decoder.
  - We performed **incremental ablation** experiments on each proposed module, corresponding to Tab. 3, showcasing the effectiveness of the proposed method.
- **Q3:** We elaborated on the **limitations of the method**, noting that it is not lightweight enough for high-resolution anomaly detection tasks. Additionally, the current Mamba-based encoder has certain gaps when directly applied to the anomaly detection domain, as shown in the ablation experiments in Tab. 10. We will continue to investigate these issues in future research.

### Reviewer aMDx
- **Q1:** We further explored:
  - The **scanning directions**, corresponding to Tab. 2.
  - The **scanning methods**, corresponding to Tab. 5, and their sensitivity to the method.
- **Q2:** We added three **additional datasets** from different categories to demonstrate the method's effectiveness, including medical, 3D, and general scene anomaly detection datasets, with ablation experiments corresponding to Tab. 4.
- **Q3:** We discussed the **robustness of the method** under different environmental conditions.
- **Q4:** We addressed the same issue as in Reviewer FXx2's Q3 and further discussed the method's **limitations and future work**.

### Reviewer gwhg
- **Q1:** We explored **comparisons with Transformer-based methods**, with ablation experiments in Tab. 1.
- **Q2:** We conducted fair comparisons by replacing the **backbone networks** of UniAD and MambaAD with three different networks, with ablation results shown in Tabs. 7 and 8.
- **Q3:** We discussed the reasons for poor anomaly localization performance and validated the method's effectiveness in pixel-level anomaly localization using the additional **mIoU anomaly localization evaluation metric** on six datasets, as shown in Tab. 6. We also provided more qualitative analysis with the RD4AD framework in Fig. 1.

### Reviewer wfnx
- **Q1:** We delved deeper into the **motivation** of this study.
- **Q2:** We conducted ablation experiments on the **local and global branches of LSS**, as shown in Tab. 9.
- **Q3:** We addressed the same issue as in Reviewer gwhg's Q3, discussing the reasons for poor anomaly localization performance and validating the method's effectiveness in pixel-level anomaly localization using the additional **mIoU anomaly localization evaluation metric** on six datasets, as shown in Tab. 6. We also provided more qualitative analysis with the RD4AD framework in Fig. 1.
- **Q4:** We conducted additional ablation experiments on the **Mamba-based encoder**, as shown in Tab. 10.
- **Q5:** We addressed the same issue as in Reviewer aMDx's Q1, with in-depth exploration and ablation experiments on the **scanning method**, as shown in Tab. 5.

---

> ### Comment · Reviewer_FXx2 · 2024-08-11
> **Question on the Uni-Medical dataset**
>
> Thank you for the clarification. I noticed that the authors referenced [46] for the Uni-Medical dataset. However, the paper in [46] neither includes experiments on medical anomaly detection nor references any medical anomaly dataset. Could the authors please confirm the accuracy of this citation for the Uni-Medical dataset?

---

> > ### Author Response · Authors · 2024-08-12
> > **Uni-Medical dataset reference.**
> >
> > Thank you for your response. The file **./data/README.md in the official code directory** provided in the abstract of paper [46] documents that **Uni-Medical originates from [A]**. After extracting the Uni-Medical dataset and comparing it with the BMAD dataset proposed in [A], we found that the authors of **[46] selected 3 out of the 6 medical anomaly detection datasets mentioned in [A]. These selected datasets contain both pixel-level and image-level ground truth and were formatted to conform to the MVTec-AD dataset standard**. We have also identified this issue and *attempted to contact the authors of [46] via email to point out the discrepancies between the code and the paper*. **In the revised version, we will cite both [46] and [A] for the Uni-Medical dataset**. Once again, thank you for your efforts in improving the quality of the manuscript.
> >
> > [A] Bmad: Benchmarks for medical anomaly detection. Bao *et al.*. CVPRW’24.

---

> ### Author Response · Authors · 2024-08-13
> **Any more comments or concerns?**
>
> Dear reviewers,
>
> Thanks a lot for your effort in reviewing this submission! We have tried our best to address all the mentioned concerns/problems in the rebuttal. Feel free to let us know if there is anything unclear or so. We are happy to clarify them.
>
> Best regards!
>
> Authors of MambaAD.

---

### Decision · Program_Chairs · 2024-09-25

**Decision:**

Accept (poster)

**Comment:**

This paper proposed a Mamba-based autoencoder approach for multi-class anomaly detection. The main novelty is adapting the Mamba architecture for anomaly detection. All reviewers recognized this, as well as the extensive experiments demonstrating improved performance over CNN architectures. The paper is also well written and contains attractive diagrams. All reviewers recommended acceptance.

The AC is concerned the paper did not compare to PatchCore and other standard AD approaches. Indeed, there are some doubts if this approach outperforms standard AD approaches. The lack of comparison is an issue with many multi-class/unified AD works. Still, given the reviewer consensus and as this was not raised during the discussion, the AC gives the paper the benefit of the doubt and recommends acceptance. Please include a comparison with SoTA one-class methods in the camera-ready version.